

# The open ocean sensible heat flux and its significance for Arctic boundary layer mixing during early fall

Manisha Ganeshan[1], Dong L. Wu[2]

[1]Goddard Earth Sciences Technology and Research Studies and Investigations, USRA, Greenbelt, 20771, U.S.A.
[2]NASA Goddard Space Flight Center, Greenbelt, 20771, U.S.A.

*Correspondence to*: Manisha Ganeshan (mganeshan@usra.edu)

**Abstract.** The increasing ice-free area during late summer has transformed the Arctic to a climate system with more dynamic boundary layer clouds and seasonal sea ice growth. The open ocean sensible heat flux, a crucial mechanism of excessive ocean heat loss to the atmosphere during the fall freeze season, is speculated to play an important role in the recently observed cloud cover increase and boundary layer (BL) instability. However, lack of observations and understanding of the resilience of the proposed mechanisms, especially in relation to meteorological and interannual variability, has left a poorly constrained BL parameterization scheme in Arctic climate models. In this study, we use multi-year Japanese cruise ship observations from R/V Mirai over the open Arctic Ocean to characterize the surface sensible heat flux (SSHF) during early fall and investigate its contribution to BL turbulence. It is found that surface-generated turbulent mixing is favored during episodes of high wind speed, and is also influenced by the prevailing cloud regime. The maximum ocean-atmosphere temperature difference is observed during cold air advection (associated with the stratocumulus regime). Yet, contrary to previous speculation, the efficiency of turbulent heat exchange is low. The SSHF contribution to BL mixing is significant during the uplift (low-pressure) followed by the highly stable (stratus cloud) regime. Overall, the open ocean sensible heat flux can explain ~10% of the BL height variability, whereas mechanisms such as cloud-driven turbulence appear to be dominant. Nevertheless, there is strong interannual variability in the strength of the ocean-atmosphere coupling. The changing occurrence of Arctic climate patterns, such as positive surface wind speed anomalies, can easily enhance the ocean's contribution to BL turbulence. This study highlights the need for comprehensive boundary layer observations such as the R/V Mirai for better understanding and predicting the dynamic nature of the Arctic climate.





## 1 Introduction

The recent decline of the Arctic sea ice during late summer (Aug-Sep) has raised several questions for the new climate system, for example, the response of boundary layer clouds and the feedback to sea ice recovery. Turbulent heat fluxes over the ice-free ocean are expected to play an important role under these circumstances. The aim of this study is to provide a

5 better understanding of the role of surface sensible heat flux in the formation of oceanic boundary layer and dissipating ocean heat to the atmosphere during late summer and early fall.

Model simulations of the 21$^{st}$ century climate have suggested that even if the Arctic Ocean were to become completely ice-free in summer, the loss of excess heat to the atmosphere through enhanced ocean (sensible and latent) heat fluxes during October-December months would enable the recovery of sea ice (Tietsche et al. 2011). Furthermore, ship-

10 based observations during the fall of 2010 have also indicated the importance of the ocean sensible heat flux for the onset of the annual freeze cycle (Inoue and Hori 2011). The authors suggested that the cold air outbreak in the wake of cyclogenesis enabled significant cooling of the upper ocean (and freeze onset; Inoue and Hori 2011). Presently, however, there is limited observational guidance for the open ocean sensible heat flux and the efficiency of turbulent heat exchange during such events.

The rapid sea-ice retreat in recent years has also raised speculation that the increased air-sea temperature gradients may contribute to reduced boundary layer stability and associated cloud changes (Kay and Gettelman 2009; Schweiger et al. 2008). For example, observations from satellites (Kay and Gettelman 2009; Wu and Lee 2012;) and ground stations (Eastman and Warren 2010) suggest a general increase in the low cloud cover over the ice-free Arctic Ocean during fall, especially in regions of reduced atmospheric stability (Kay and Gettelman 2009). Studies have also shown an increase in the

mid-level cloud cover (and simultaneous decrease in low clouds) indicating a deepening of the Arctic boundary layer (Sato et al. 2012; Schweiger et al. 2008; Palm et al. 2010). This has also been attributed mainly to the enhanced air-sea temperature difference and resulting upward sensible heat flux (Sato et al. 2012; Schweiger et al. 2008). Despite the observed cloud changes, no direct measurements have been made to quantify the open ocean surface fluxes or its influence on boundary layer mixing. As this area continues to increase in a warmer climate, it becomes more important to fully

understand and characterize the changes in cloud cover and the underlying boundary layer processes.

Observations over sea ice show that mixing in the Arctic boundary layer is primarily driven by cloud top radiative cooling (Tjernstrom et al. 2004; Inoue et al. 2005; Morrison et al. 2011; Shupe et al. 2013), which is often decoupled from surface turbulent fluxes (Curry et al. 2000; Morrison et al. 2012; Shupe et al. 2013; Nicholls and Leighton 1986). Such a BL represents the situation arising from strongly insulating sea ice that prevents efficient turbulent heat exchange at the surface.

Over the open ocean, the air-sea interaction can be more pronounced. For example, it has been suggested that surface heat and moisture fluxes lead to boundary layer "roll" clouds during cold air outbreak events over the open Arctic Ocean (Klein et al. 2009). In this study, using multi-year ship-based observations, we investigate the variability of the open ocean sensible



heat flux, and more importantly its contribution to boundary layer mixing (and turbulent heat exchange) under varying weather (and cloud) regimes.

## 2 Datasets

Surface and upper-air meteorological data from the ice-strengthened research vessel (R/V Mirai) operated by the Japan
Agency for Marine-Earth Science and Technology are analyzed in this study. The vessel surveyed the ice-free regions mainly in the vicinity of Beaufort and Chukchi seas (from 125° W - 175° E longitude and between 60° - 80° N latitude). Data were collected during September and early October of the years 2002 (Fujiyoshi and Shimada 2002), 2004 (Fujiyoshi and Shimada 2004), 2008 (Kurita and Yoneyama 2008), 2009 (Inoue and Yoneyama 2009), 2010 (Inoue 2010) and 2013 (Inoue 2013). On account of the retreating sea ice, recent observations were collected in more northern latitudes compared to
earlier years (Fig. 1; Sato et al. 2012; Inoue and Hori 2011). The year 2013, however, is an exception when observations were primarily collected at a fixed point (72.75 °N and 168.25 °W) as part of the Arctic Research Collaboration for the Radiosonde Observing System (ARCROSE) experiment (Inoue et al. 2015; Kawaguchi et al. 2015). The radiosonde data include profiles of temperature, pressure, winds, and relative humidity, typically sampled at 3 to 12 hour intervals with vertical resolution ranging from 40 m in the lower levels to around 70 m in the mid-troposphere. In order to exclude the
near-surface contamination due to warming and cooling of the ship body, a minimum height threshold of 100 m is imposed to ensure quality data of the atmospheric profiles.

Also used in the R/V Mirai data analysis are independent, quality controlled observations of surface meteorological variables, viz., 10-minute average values of sea surface temperature (SST), surface air temperature (SAT), surface pressure, and surface horizontal wind speed ($V_{surf}$). The SAT and $V_{surf}$ are measured at 21 m and 25 m above sea level, respectively.
These measurements are used to estimate the surface sensible heat flux (SSHF) at the time of radiosonde launches. A total of 876 contemporaneous samples of boundary layer structure and SSHF (excluding missing data) are analyzed from all cruises. Thus, this dataset provides a unique and valuable survey of Arctic boundary layer properties over the open ocean (Fig. 1), complementing other observational efforts carried out mainly on the central polar ice pack and coastal/continental Arctic regions.

## 25   3 Methods

### 3.1 Estimation of surface sensible heat flux

The product of the wind speed, and the temperature difference between the sea surface and overlying air (ΔT), is a common estimate of the SSHF (Fairall et al. 1996; Bourassa et al. 2010; Inoue et al. 2011). In this study, the SSHF is calculated using the following equation,





$$SSHF \quad = \quad \rho C_p C_H V_{surf} \, (\varDelta T), \tag{1}$$

where $\varDelta T = SST - SAT$, °C,

$\quad\quad V_{surf}$ = surface horizontal wind speed, m s$^{-1}$

$\quad\quad \rho =$ air density, kg m$^{-3}$

$\quad\quad C_p =$ specific heat capacity of air, J kg$^{-1}$ K$^{-1}$

$\quad\quad C_H =$ transfer coefficient for SSHF (based on winds measured at 20 m height)

The values of the constants $\rho$, $C_p$, and $C_H$, are adopted from Inoue et al. (2011). Based on Eq. 1, positive values of SSHF will

indicate upward surface sensible heat flux (from the ocean to the atmosphere).

**3.2 Determination of boundary layer height**

The height of the well-mixed boundary layer is calculated using the parcel-based method, as illustrated in Fig. 2. In this method, a surface parcel is assumed to ascend along a dry adiabat up to the lifting condensation level (LCL), and along a moist adiabat thereafter (denoted by red dashed line in Fig. 2). There is no striking difference in the slopes of the dry and

15 moist adiabats in Fig. 2, as they are nearly parallel in a cold environment. The height of the boundary layer (BL) is computed as the level where the parcel temperature falls below the environment temperature by a value of 0.6 °K or more. For most soundings, this level is coincident with the base of the temperature inversion (as shown in Fig. 2).

As observed by Sato et al. (2012), the boundary layer over the open Arctic Ocean appears to be well-mixed. We further inspect the wind speed variance in the BL, which is typically less than 30% of the mean, suggesting negligible

mechanical turbulence for most cases. The Arctic BLs appear to be primarily mixed by convective fluxes. In a well-mixed BL, the surface turbulent fluxes decrease linearly with height (Holton 2004). In order to estimate the contribution of surface fluxes to boundary layer mixing, we correlate the SSHF with the BL height on a profile-by-profile basis. A good correlation implies that the surface fluxes control the convective turbulence in the BL. In the case of weak correlation, factors such as convective mixing from cloud-generated turbulence are to be considered (Morrison et al. 2012; Shupe et al. 2013). To

comprehend the complex BL processes, we also examine the cloud layer thickness within the BL as described in the following section.

**3.3 Estimation of boundary layer cloud thickness ratio**

Based on past studies over sea ice, the Arctic is found to be mostly cloudy, and the cloud-driven turbulence is known to control the BL height variability (Tjernstrom et al. 2004; Shupe et al. 2013). Low clouds at the top of the boundary layer

may generate convective turbulence from above. Such a boundary layer can be decoupled from the surface (Curry et al. 2000; Morrison et al. 2012; Shupe et al. 2013), which may result in a poor correlation between the SSHF and BL height. It is therefore important to consider the effect of boundary layer clouds in this study. For this purpose, the BL cloud thickness





ratio is calculated as the percentage ratio of cloud layer within the BL. The cloud-base is defined as the first layer above the surface where the relative humidity (RH) equals 90% or more (as in Sato et al. 2012), and the cloud layer is calculated as the vertical integral of all layers within the boundary layer that exceed 90% RH. As an example, for the profile shown in Fig. 2 (inset), the cloud-base and BL heights are calculated as 580 m and 1180 m respectively, and the BL cloud thickness ratio is estimated to be ~50%.

## 4 Results

### 4.1 Characteristics of SSHF over the open Arctic Ocean

Figure 3 shows the distribution of the SSHF, the temperature gradient between sea surface and air ($\Delta T$), and the surface wind speeds ($V_{surf}$), based on multi-year observations. The lack of meaningful differences in the yearly SSHF median values suggests that its interannual variability is not significant at the 95% confidence level. The correlation between SSHF and $\Delta T$ (*correlation coefficient* = 0.77) is found to be higher than that of SSHF and $V_{surf}$ (*correlation coefficient* = 0.45). The distribution in Fig. 3 (a) indicates that more positive SSHF values, and a heavier tail (barring outliers), occur in recent years (2009, 2010, 2013). It appears that an increased SAT variability (not shown) may contribute to some extent to the broader $\Delta T$ and SSHF distribution observed during 2009 and 2010 (Figs. 3 (a) and (b)). (In general, the $\Delta T$ appears to be more strongly influenced by SAT rather than SST).

Thus, it appears that the SSHF can be sensitive to the $\Delta T$ variability during fall, particularly to negative SATs. In model simulations, the occurrence of cold air advection (CAA) events and increased $\Delta T$, is known to release copious amounts of ocean heat flux and trigger the seasonal recovery of sea ice (Kolstad and Bracegirdle 2008). Models also project that future occurrences of CAA may spread further poleward along the retreating sea ice margin (Kolstad and Bracegirdle 2008). In spite of recent observations and modeling efforts (Inoue and Hori 2011; Klein et al. 2009), there are limited measurements of actual surface fluxes during such events. In the following sections, using ship-based measurements, we investigate the instantaneous relationship between the surface sensible heat flux and the boundary layer height, which qualitatively represents the efficiency of turbulent heat exchange between the ocean and the atmosphere.

### 4.2 The SSHF contribution to boundary layer mixing

Figure 4 (a) shows that a weak positive relationship exists between the SSHF and BL height, which can explain up to 10% of the BL height variability. Contrary to expectations, this relationship is not found to depend on the ocean-atmosphere temperature gradient ($\Delta T$). Instead, the correlation coefficient ($r$) is sensitive to the surface wind speeds ($V_{surf}$; Fig. 4 (b)) suggesting that the surface-generated turbulent mixing is favored during episodes of strong winds. Note that although $\Delta T$ does not appear to contribute to the instantaneous ocean-atmosphere turbulent heat exchange, it may influence the SSHF on longer (diurnal, seasonal, and interannual) time-scales.





Additionally, low-level clouds are known to generate turbulence in the Arctic BL and may influence the correlation. Therefore, we closely inspect the behavior of the correlation coefficient (*r*) under varying cloud regimes in the following subsection.

### 4.2.1 Effects of cloud regime

In the Arctic, large-scale atmospheric processes are largely responsible for the occurrence and sustenance of low cloud cover (Herman and Goody 1976; Morrison et al., 2012; Solomon et al., 2014). Barton et al. (2012) recently classified Arctic clouds based on the background dynamic and thermodynamic state of the lower troposphere. They identified four robust meteorological regimes based on the lower tropospheric stability or the potential temperature difference between the surface and 700 mb ($\theta_{700}$ - $\theta_{surf}$), and the 500 mb pressure vertical velocity ($\omega_{500}$). The first three regimes have positive $\omega_{500}$ values

indicating weak subsidence, and differ only in their lower tropospheric stability ($\theta_{700}$ - $\theta_{surf}$; Barton et al. 2012). The fourth atmospheric state comprises the uplift regime (characterized by rising motion or negative $\omega_{500}$ values), which is found to occur 10-15% of the time suggesting that the Arctic troposphere is dominated by weak subsidence (Barton et al. 2012; Taylor et al. 2015). The subsidence regimes typically have cloud bases within the boundary layer, and are characterized by increasing cloud-top/BL height with decreasing stability (Barton et al. 2012). During the uplift regime, on the other hand, the

cloud fraction peaks in the free troposphere above the boundary layer (Barton et al. 2012). In our study, the focus is on the boundary layer height variability therefore we will not be studying clouds with cloud-base above the BL. Moreover, in lieu of the pressure vertical velocity ($\omega_{500}$), we examine the surface pressure to distinguish between the uplift and subsidence regimes.

The bottom panel of Figure 5 (a) shows the frequency distribution of BL cloud thickness ratio during fall. Three

distinct BL cloud types emerge, with cloud ratios peaking at 5%, 65%, and 95% (Fig. 5 (a)). Consequently, the observations are divided into three groups consisting of low (<20%), medium (20-80%), and high (>80%) BL cloud thickness ratios, respectively. Note that threshold pairs other than 20-80% (such as 10-90% and 25-75%) were tested for classification purposes, and the results were found to remain robust. Table 1 shows the occurrence frequency, mean surface pressure, and the lower tropospheric stability ($\theta_{700}$ - $\theta_{surf}$) associated with each group, while Figs. 5 (b)-(d) show the average temperature

and moisture profiles. Significantly lower pressure conditions (99% confidence level) are associated with the low BL cloud thickness group, which occurs roughly 15% of the time. While upper level clouds may be present, a vast majority of the cases do not have boundary layer clouds (Fig. 5 (b)). Therefore this group is synonymous with the uplift regime described by Barton et al. (2012). On the other hand, the top panel of Figure 5 (a) suggests that BL clouds are favored in a subsiding environment (high-pressure conditions), consistent with Barton et al. (2012).

For the group with high cloud thickness (greater than 80%), the relatively strong lower tropospheric stability (Table 1) and shallow BL height (Fig. 5 (d)) indicate that it belongs to the very highly stable/highly stable regime described by Barton et al. (2012). Figure 5 (c) on the other hand shows a deeper boundary layer with moderate stability suggesting that the group with medium cloud thickness (20 to 80%) is similar to the stable regime described by Barton et al. (2012). Note that



the occurrence frequency of these two groups (Table 1) also aligns with that of the respective stable regimes observed during fall (Barton et al. 2012).

In fact, a comparison of the lower tropospheric stability and BL (cloud-top) height between both groups (Table 1 and Figs. 5 (c), (d)) reveal that they are in fact analogous to the stratocumulus and stratus cloud types. The stable, shallow, and cloudy boundary layer in Fig. 5 (d) is characteristic of the stratus cloud regime, whereas the deeper well-mixed BL with higher cloud top (Fig. 5 (c)) represents the stratocumulus-topped boundary layer. Similar distinctions between stratus and stratocumulus Arctic clouds were noted in previous studies as well (Sato et al. 2012). Thus, the low, medium, and high cloud thickness groups identified in Table 1, are henceforth referred to as the uplift, stratocumulus, and stratus regimes, respectively. Figure 6 compares the lower tropospheric structure of temperature and moisture for stratus and stratocumulus regimes. Consistent with Sato et al. (2012), it is evident that CAA is mainly responsible for the occurrence of stratocumulus clouds (Fig. 6 (a)) whereas warm and moist air advection (or subsidence) leads to the formation of stratus clouds in the Arctic (Fig. 6).

Table 1 shows the average wind speed, $\Delta T$, and the correlation coefficient between SSHF and BL height ($r$), for the three regimes. The wind speeds are comparable, but the $\Delta T$ is significantly higher for the stratocumulus regime (99% confidence level). In the past, there has been speculation that the surface-generated turbulence is enhanced due to strong air-sea temperature gradients and reduced stability associated with cold air advection over open water (Sato et al. 2012; Kay and Gettelman 2009). Yet, the weak correlation coefficient ($r$) for the stratocumulus regime suggests that despite the reduced stability and enhanced $\Delta T$, the surface-driven turbulence remains insignificant (Table 1). Fig. 6 (a) shows that the temperature anomaly in this regime is maximized between 300 to 1.5 km altitudes, indicating that stratocumulus clouds likely form by saturating to the significantly colder air mass that is advected *above* the surface. This could lead to an unstable lapse rate fueled by saturation and the release of latent heat flux. Thus, processes decoupled from the surface are likely driving the boundary layer height variability in the stratocumulus regime.

On the other hand, the surface contribution to boundary layer mixing is significant during the uplift regime, as well as in the presence of stratus clouds occurring within a (warm and wet) subsiding environment ($r$ in Table 1). Other studies have also noted that the influence of surface type (sea ice vs. open water) is more significant for shallow boundary layer clouds occurring in the highly stable (stratus) regime compared to the stable (stratocumulus) regime (Barton et al. 2012; Taylor et al. 2015). In the following subsection, we will more closely examine the relationship between the SSHF and BL height in the uplift and stratus regimes.

### 4.2.2 Influence of surface winds

Figure 7 (a) shows that the SSHF can explain a substantial amount of the BL height variability in the uplift regime (up to 37%). For the stratus cloud regime (Fig. 7 (b)), the correlation between SSHF and BL height is significant but improves substantially during episodes of high surface wind speed (> 9.8 ms$^{-1}$). For the deep, stratocumulus-topped boundary layer,





the relationship between SSHF and BL height becomes weakly positive ($r = 0.14$) during high surface wind speeds but remains insignificant. Thus, apart from other factors, surface winds are clearly important for generating turbulent heat exchange in the stable Arctic boundary layer.

Altogether, based on the linear relationships between SSHF and BL height and the frequency of occurrence of the

uplift (~15%) and stratus cloud regimes (~41%), the surface-generated turbulence may explain up to 10% of the Arctic BL height variability during fall. Whereas the in-cloud moist and radiative processes that are responsible for BL mixing over sea ice (Tjernstrom et al. 2004; Shupe et al. 2013; Morrison et al. 2012) are likely to be dominant over the open ocean as well. In the following subsection, we examine the interannual variability in the BL height and its relationship to the SSHF.

### 4.3 Interannual variability

As discussed in the previous section (4.2.3), the height of the well-mixed Arctic BL is likely controlled by the cloud-generated turbulence more than the SSHF. Figure 8 shows the yearly distribution of BL height, BL cloud thickness ratio, and surface pressure, for the period of the cruise. Compared to the SSHF (Fig. 3 (a)), the BL height has more interannual variability (Fig. 8 (a)) as suggested by the significantly shallow boundary layer observed during 2002 (95% confidence level). The large-scale circulation appears to be different during this year as indicated by the anomalous sea level pressure

distribution in Fig. 8 (c). The circulation anomaly likely has a significant influence on the BL cloud distribution (Fig. 8 (b)), which consequently has an impact on the BL height variability (Fig. 8 (a)).

The interannual variability in the correlation coefficient ($r$; Table 1) suggests that it is most significant for the years 2002 and 2010 (99% confidence level). The time-series of the SSHF along with BL height during these two years, indeed confirms its dominant role in generating turbulence (Fig. 9). The reasons for the same are explored below.

As described in section 4.2, both cloud-type and wind speeds may influence the SSHF contribution to BL mixing. The frequency distribution of different cloud regimes and the corresponding surface wind speed anomalies (positive only) are shown for each year in Fig. 10. For the years with positive Arctic Oscillation (Table 2), it is evident that the stratus and stratocumulus regimes dominate the climate. The regime distribution is quite different during 2002 and 2013 (both years with negative AO index). The year 2002 is governed by anomalously low surface pressure or the uplift regime, whereas the

25  year 2013 is accompanied by greater than usual occurrence of stratocumulus clouds (bottom panel of Fig. 10). Sampling inconsistency due to spatially restricted (fixed point) observations can also contribute to the anomalous cloud regime distribution observed during 2013. From Fig.10, it appears that a lower stratocumulus cloud fraction (bottom panel) coupled with higher surface wind speeds in the stratus regime (top panel), contribute to the better correlation between SSHF and BL height during 2002 and 2010 (Table 2).



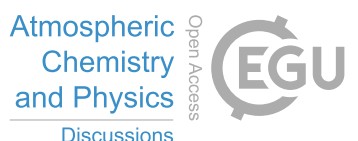

Thus, although the SSHF presently explains only 10% of the BL height variability, it has a strong control on the BL mixing during certain years and certain regimes (especially in the presence of high surface wind speeds). As a result, the changing occurrence of Arctic climate patterns in the future, such as stronger winds, may easily enhance the surface contribution to BL turbulence.

## 5 Discussions

### 5.1 SSHF feedbacks to freeze onset and sea ice recovery

In the open Arctic Ocean, surface turbulent heat fluxes are an important process for the loss of absorbed heat (during summer) to the atmosphere. A recent modeling experiment has shown that the sensible and latent heat fluxes during fall and early winter are crucial for the recovery of sea ice in the future ice-less Arctic (Tietshce et al. 2011). In this study, we investigate the SSHF variability during early fall as well as its role in BL mixing using multi-year ship-based meteorological observations. Based on its contribution to boundary layer turbulence, we are able to estimate the efficiency of turbulent heat exchange between the open ocean and the atmosphere under varying weather (and cloud) regimes.

The SSHF over the open Arctic Ocean is mostly positive during fall (~90% of the time) owing to the positive ocean-atmosphere temperature differences ($\Delta T$). In agreement with past studies (Sato et al. 2012; Klein et al. 2009), the highest $\Delta T$ is observed during cold air advection events that are associated with the stratocumulus cloud regime (Table 1). This regime occurs around 44% of the time as indicated in Table 1. It has been speculated that the increased $\Delta T$ and associated sensible heat flux is responsible for generating turbulence and reducing the stability of the stratocumulus-topped BL (Sato et al. 2012; Kay and Gettelman 2009). However, based on the weak correlation between SSHF and BL height in Table 1, this does not appear to be true. In fact, the variability of the correlation coefficient ($r$) as a function of winds (Fig. 4 (b)), suggests that the efficiency of turbulent heat exchange between the open ocean and atmosphere is more sensitive to surface wind speeds than $\Delta T$. (Nevertheless, $\Delta T$ can have a possible influence on the SSHF on diurnal, seasonal, and interannual time-scales).

Inoue and Hori (2011) suggested that CAA in the wake of cyclogenesis may contribute substantially to oceanic heat loss through SSHF, and trigger the onset of freeze events. From this study, it appears that CAA alone may not produce substantial heat dissipation, but when accompanied by strong winds it can lead to significant upper-ocean cooling and freeze





onset. It is therefore important to monitor the frequency of both CAA and surface wind speeds, as they can be crucial for the recovery of the rapidly declining Arctic sea ice. While the former can lead to maximum $\Delta T$ values that may enhance SSHF, the latter appears to be more efficient in dissipating the ocean heat to the atmosphere.

## 5.2 SSHF control on boundary layer height

Previous BL observations over the frozen Arctic Ocean have suggested that in-cloud radiative and moist thermodynamic processes are the most significant factors for BL turbulent mixing especially since surface fluxes are negligible (Tjernstrom et al. 2004; Shupe et al. 2013; Curry et al. 2000). Over the ice-free Ocean, however, surface fluxes are expected to be important due to the lack of insulating sea ice. Recent observations of increased cloud cover and reduced BL stability during fall, have led to speculation that the surface-driven turbulence is enhanced over the open Arctic Ocean due to the intensified

air-sea temperature gradients (Kay and Gettelman 2009; Schweiger et al. 2008; Sato et al. 2012). In this study, the BL in the same region is found to be well-mixed, with the SSHF controlling up to 10% of its height variability (as explained in section 4.2.2). Thus, other factors such as cloud-driven turbulence are likely to be dominant, even over the open Arctic Ocean. Nevertheless, the surface influence on BL mixing is particularly pronounced under certain weather (and cloud) regimes and during certain years (for example, 2002 and 2010).

The largely positive SSHF during fall is the most efficient in generating BL turbulence under the uplift (low-pressure) regime which typically occurs ~15% of the time. During most years, a high-pressure (subsidence) regime is observed, dominated by boundary layer stratus and stratocumulus clouds (Table 1; Figs. 8 and 10). In agreement with previous studies (e.g., Sato et al. 2012), the troposphere is found to be warm and moist in the very stable stratus regime characterized by a fully saturated, shallow boundary layer (Fig. 5 (d) and Fig. 6). In this regime, the SSHF contributes

significantly to BL mixing, especially under strong wind conditions (Fig. 7 (b)). (Note that the interannual variability in the relationship between SSHF and BL height is sensitive to the wind speed anomalies observed during this regime, as seen in section 4.3).

On the other hand, although associated with a more unstable lapse rate and a higher cloud base, there is no significant surface contribution to BL mixing in the stratocumulus regime (Fig. 5 (c)). It has been speculated that the

instability in this regime is produced by increased air-sea temperature gradients (Sato et al. 2012; Kay and Gettelman 2009).



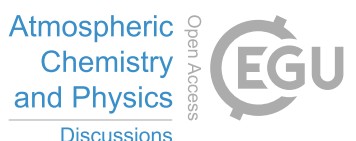

Our results suggest that despite significantly positive ∆T values, there appears to be a low efficiency in the instantaneous turbulent heat exchange between the ocean and the atmosphere (Table 1). Other studies have also noted that the influence of surface type (sea ice vs. open water) is more significant for the shallow BL clouds occurring in the highly stable (stratus) regime compared to the less stable (stratocumulus) regime (Barton et al. 2012; Taylor et al. 2015). It is possible that cloud condensation processes (and latent heat fluxes) in the cold saturated air are responsible for the formation of the deep, well-mixed BL observed during the stratocumulus regime (Fig. 6).

In summary, the SSHF appears to contribute little to turbulent mixing in the stratocumulus-topped BL, but can significantly influence the BL height during the uplift and the stratus cloud regimes (especially in the presence of strong surface winds).

## 6 Conclusions

For the rapidly evolving Arctic region, a comprehensive understanding of ocean-atmosphere interactions and underlying coupling processes is crucial for improving regional and global climate models and their predictability. Current models have significant differences in important physical processes such as the efficiency of turbulent heat transfer from the ocean to the atmosphere (Kolstad and Bracegirdle 2008; Grønas and Skeie 1999; Pagowski and Moore 2001). The primary goal of this study was to evaluate and quantify the role of open ocean sensible heat flux in Arctic BL turbulence based on multi-year ship-based observations acquired during early fall. The main conclusions are summarized as follows:

- The surface sensible heat flux during fall is mostly positive owing to the positive ocean-atmosphere temperature differences over the ice-free ocean. Yet, the instantaneous atmospheric response to the enhanced fluxes only occurs during specific weather (and cloud) regimes. For example, cold air advection often leads to larger air-sea temperature gradients, but the dissipation of the ocean heat through SSHF is mainly favored during high surface wind speed episodes.

- Consistent with previous studies (Tjernstrom et al. 2004; Shupe et al. 2013; Morrison et al. 2012), it is evident that the large-scale circulation strongly affects the BL height variability in the Arctic through its dynamical influence on BL clouds (Fig. 8). Fig. 10 shows that both stratus and stratocumulus clouds are frequently observed in the boundary layer over the open Arctic Ocean, prevalent roughly 85% of the time. (The year 2002 is an exception with low BL cloud fraction due to the anomalously low surface pressure conditions). In agreement with previous work (Sato et al. 2012), it is found that stratus clouds are often associated with warm air advection (and high stability) whereas stratocumulus clouds result from CAA (and lower stability). However, contrary to speculation, the surface generated turbulence is more strongly favored in the former compared to the latter. The decreased stability





associated with the stratocumulus-topped BL appears not to be caused by vigorous sensible heat exchange between the warm ocean and the cold atmosphere but rather by latent heat flux release upon saturation to the cold advected air.

- Overall, the surface sensible heat flux explains only up to 10% of the BL height variability for all cases observed by R/V Mirai. Yet, there is pronounced interannual variability in this relationship due to anomalous wind and circulation patterns. In particular, the SSHF is found to be most significant for Arctic BL turbulence in the uplift regime (~ 15% of the cases), as well as under the influence of strong surface winds that occur in the warm and moist stratus regime (~ 8% of the cases). During the years 2002 and 2010, the surface-generated turbulence is found to be crucial for BL height variability, likely due to the occurrence of strong surface winds associated with the stratus regime.

This study highlights the need for comprehensive in-situ observations to improve model physics for more reliable projections of the coupled Arctic climate and sea ice in the future. Using available surface and upper-air observations from ship cruises, we provide first-hand insights of the optimal conditions for the SSHF contribution to BL turbulent mixing. Yet, the role of latent heat fluxes in BL moistening and cloud formation remains unexplored. The regional and seasonal scale variability in SSHF and BL height warrants further investigation as well, which will be pursued in future studies. Nevertheless, the relevant mechanisms identified in this study can be incorporated in climate models to investigate the changing occurrence of weather patterns and its influence on Arctic sea ice recovery/freeze onset through the efficient release of SSHF.

*Acknowledgements*

This work is supported by NASA Earth Science GNSS Remote Sensing and Interdisciplinary Research programs. Data used in this study were acquired during the MR02-K05 Leg 1, MR04-05, MR08-04, MR09-03 Leg 2, MR10-05 Leg 2, and MR13-06 Leg 1 cruises of R/V Mirai, Japan Agency for Marine-Earth Science and Technology. The Arctic Oscillation Index values were obtained from the National Oceanic and Atmospheric Administration's Climate Prediction Center using the following webpage: http://www.cpc.ncep.noaa.gov/products/precip/CWlink/daily_ao_index/ao.shtml.

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

5 **Tables**

**Table 1.** The frequency of occurrence, mean surface pressure, mean lower tropospheric stability, mean boundary layer height, cloud regime, the correlation coefficient ($r$) between SSHF and BL height, mean temperature difference between ocean and air, and mean surface wind speed, observed for the three different groups of BL cloud thickness ratio (see text and Fig. 5 for explanation). Statistically significant values of $r$ (99% confidence level) are highlighted in bold.

| BL cloud thickness ratio (%) | ≤ 20 | 20 to 80 | ≥ 80 |
|---|---|---|---|
| Occurrence frequency (%) | 15 | 44 | 41 |
| Surface Pressure (mb) | 1008 | 1015 | 1014 |
| $\theta_{700} - \theta_{surf}$ (°C) | 13.6 | 14.9 | 18.1 |
| BL height (m) | 646 | 997 | 508 |
| Cloud Regime | Uplift | Stratocumulus (stable) | Stratus (very highly/highly stable) |
| $r$ | **0.58** | -0.04 | **0.33** |
| $\Delta T$ (°C) | 1.76 | 3.41 | 1.71 |
| $V_{surf}$ (m.s$^{-1}$) | 7.01 | 7.20 | 7.05 |

**Table 2.** The average Arctic Oscillation (AO) index observed during the period of the cruise, the correlation coefficient between SSHF and BL height ($r$), and the number of observations ($n$), for each cruise year. Statistically significant values of $r$ (99% confidence level) are highlighted in bold.

| Year | 2002 | 2004 | 2008 | 2009 | 2010 | 2013 |
|---|---|---|---|---|---|---|
| AO Index | -1.16 | 0.08 | 1.21 | 0.29 | 0.51 | -1.50 |
| $r$ | **0.64** | 0.15 | 0.18 | 0.16 | **0.59** | -0.02 |
| $n$ | 100 | 65 | 93 | 131 | 214 | 273 |





**Figures**

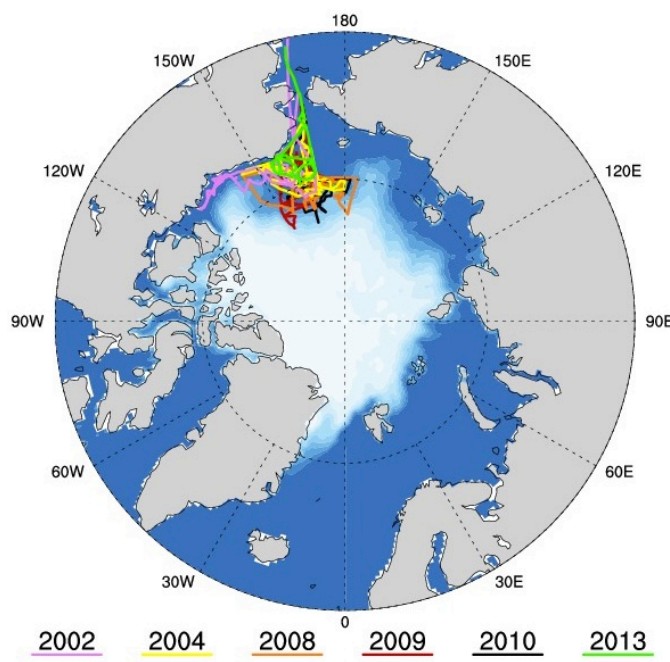

**Figure 1: Ship tracks during multi-year cruises of the R/V Mirai indicated by solid colored lines. The average ice fraction (shaded white) at the time of cruise during 2008-2010 is also shown based on National Centers for Environmental Prediction/National Center for Atmospheric Research (NCEP/NCAR) reanalyses project (Kalnay et al. 1996).**





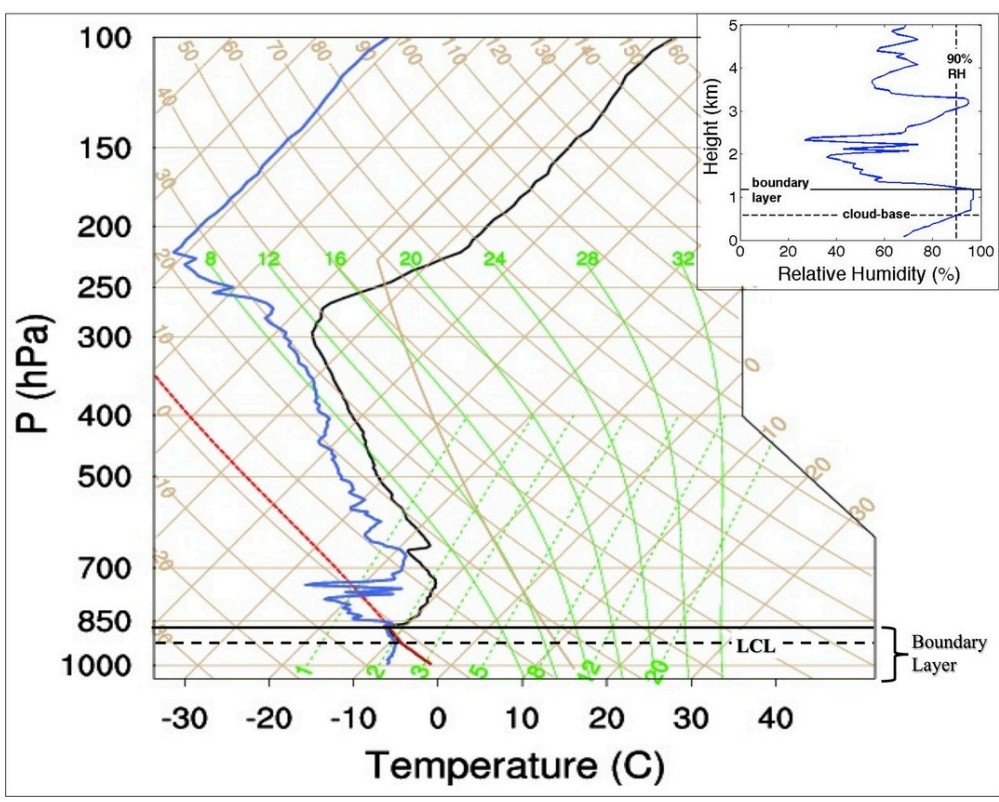

**Figure 2: Skew T- log P diagram denoting the profiles of environmental temperature (black) and dew point temperature (blue) observed at 09Z September 21, 2013. The red dashed line represents the adiabatic ascent of a surface-based parcel. The solid horizontal black line is the boundary layer (BL) height determined using the parcel-based method, whereas the dashed horizontal black line represents the lifting condensation level or LCL; and (inset) a close examination of the relative humidity in the lowest 5 km, where the dashed vertical line represents the 90% RH threshold, the dashed horizontal line is the cloud-base, and the solid horizontal line represents the boundary layer. (For detailed description on the components of a skew T-log P chart, refer to the Air Weather Service technical report titled AWS/TR-79/006,** *The Use of the Skew T, Log P Diagram in Analysis and Forecasting*, **Dec.1979, revised March 1990).**




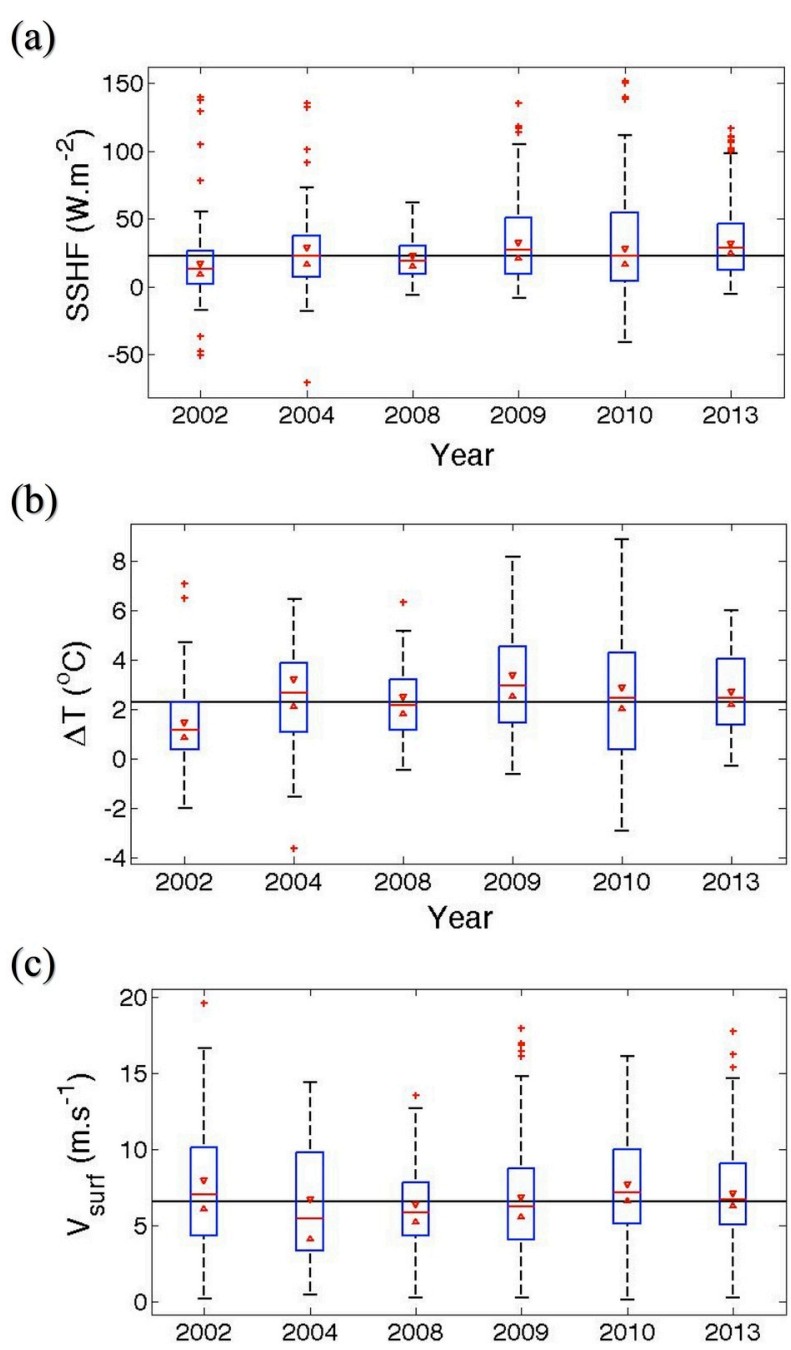

**Figure 3: The interannual variability in the distribution of (a) the surface sensible heat flux (SSHF), (b) ocean-atmosphere temperature gradient (ΔT), and (c) surface wind speeds ($V_{surf}$). The solid horizontal black line represents the overall median value based on 6 years of ship data. The median for each year is represented by the horizontal red line within each boxplot, and the red notches represent the 95% confidence intervals around the same. Two medians are different at the 5% significance level if their intervals do not overlap.**





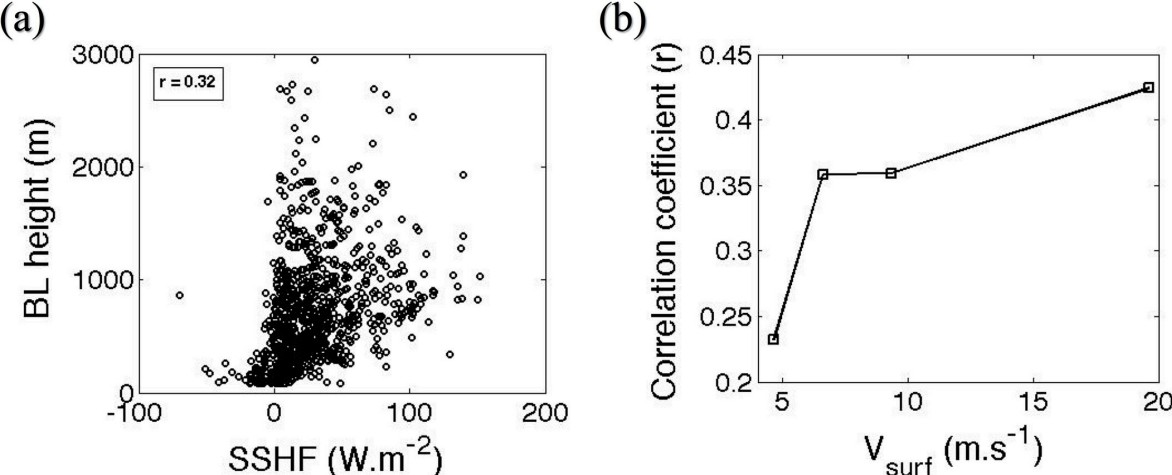

**Figure 4: (a) Scatter plot of the SSHF and BL height using all observations based on 6 years of cruise-ship data. The correlation coefficient is denoted by *r*, and (b) *r* as a function of surface wind speed ($V_{surf}$). The data are binned based on the quartiles of $V_{surf}$, and *r* is calculated for each bin. The number of observations in each bin is equal to 219.**



**Figure 5: (a)** The frequency distribution of BL cloud thickness ratio (bottom panel), and the mean surface pressure for 5 equally-spaced bins of the BL cloud thickness ratio (top panel), and **(b)** the profiles of mean temperature and dew point temperature observed for cases with BL cloud thickness ratio ≤ 20%, **(c)** same as (b) but for cases with 20% < BL cloud thickness ratio < 80%, and **(d)** same as (b) but for cases with BL cloud thickness ratio ≥ 80%.


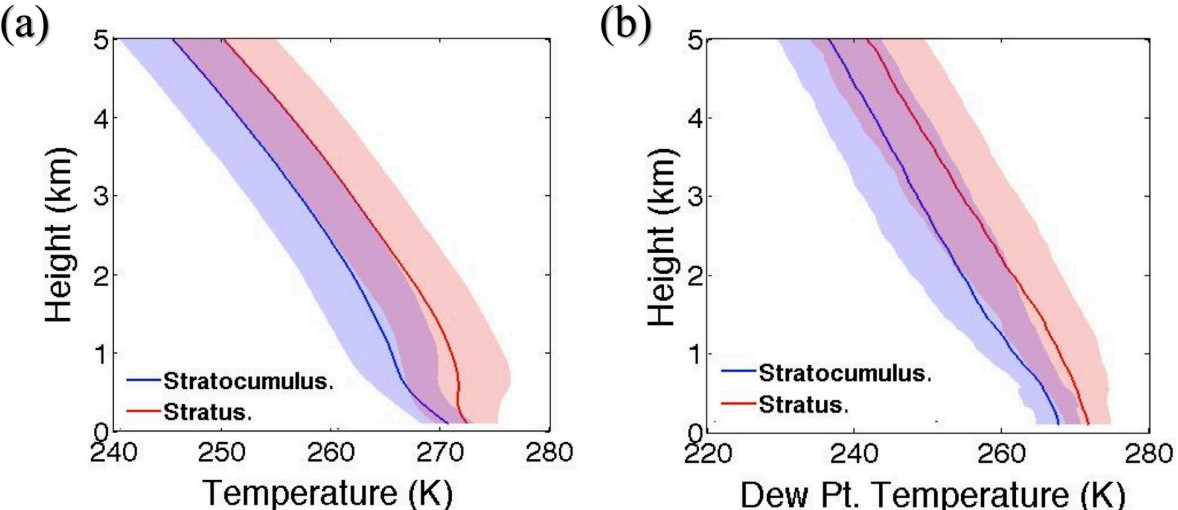

**Figure 6: The atmospheric profile comparison between the stratocumulus and stratus regimes. The solid line represents the mean and the shaded area represents the standard deviation of (a) the temperature and (b) the dew point temperature.**

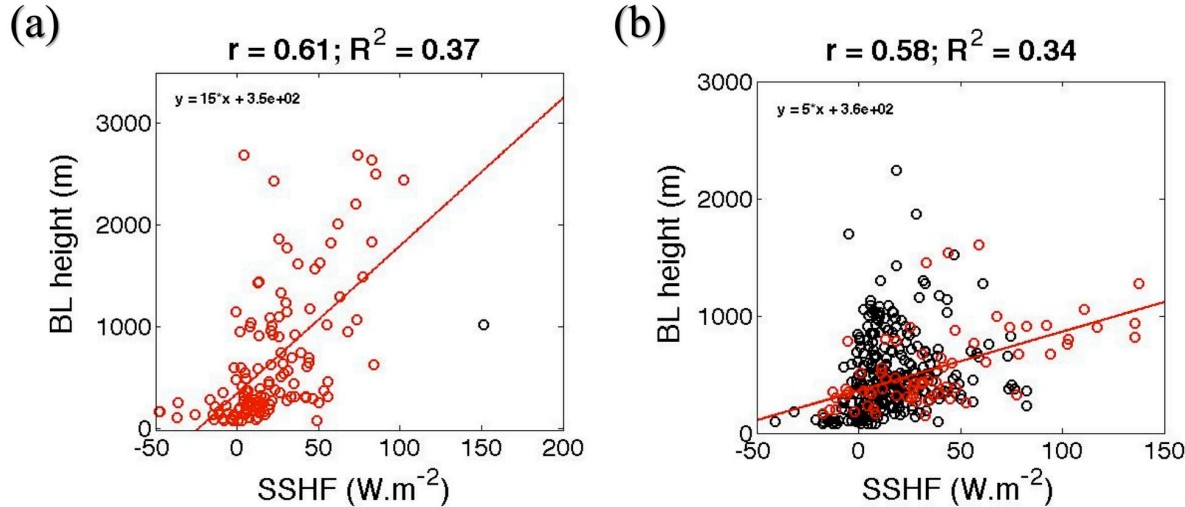

**Figure 7: The scatter plot of SSHF and BL height during (a) the uplift regime, and (b) the stratus regime. The linear relationship between the SSHF and BL height derived using the least squares method of curve-fitting for (a) all cases except one outlier (indicated by black marker), and (b) cases with surface wind speeds exceeding 9.8 ms[-1] (indicated by red markers). The correlation coefficient and the coefficient of multiple determination for each linear relationship is denoted by _r_ and _R_[2], respectively.**



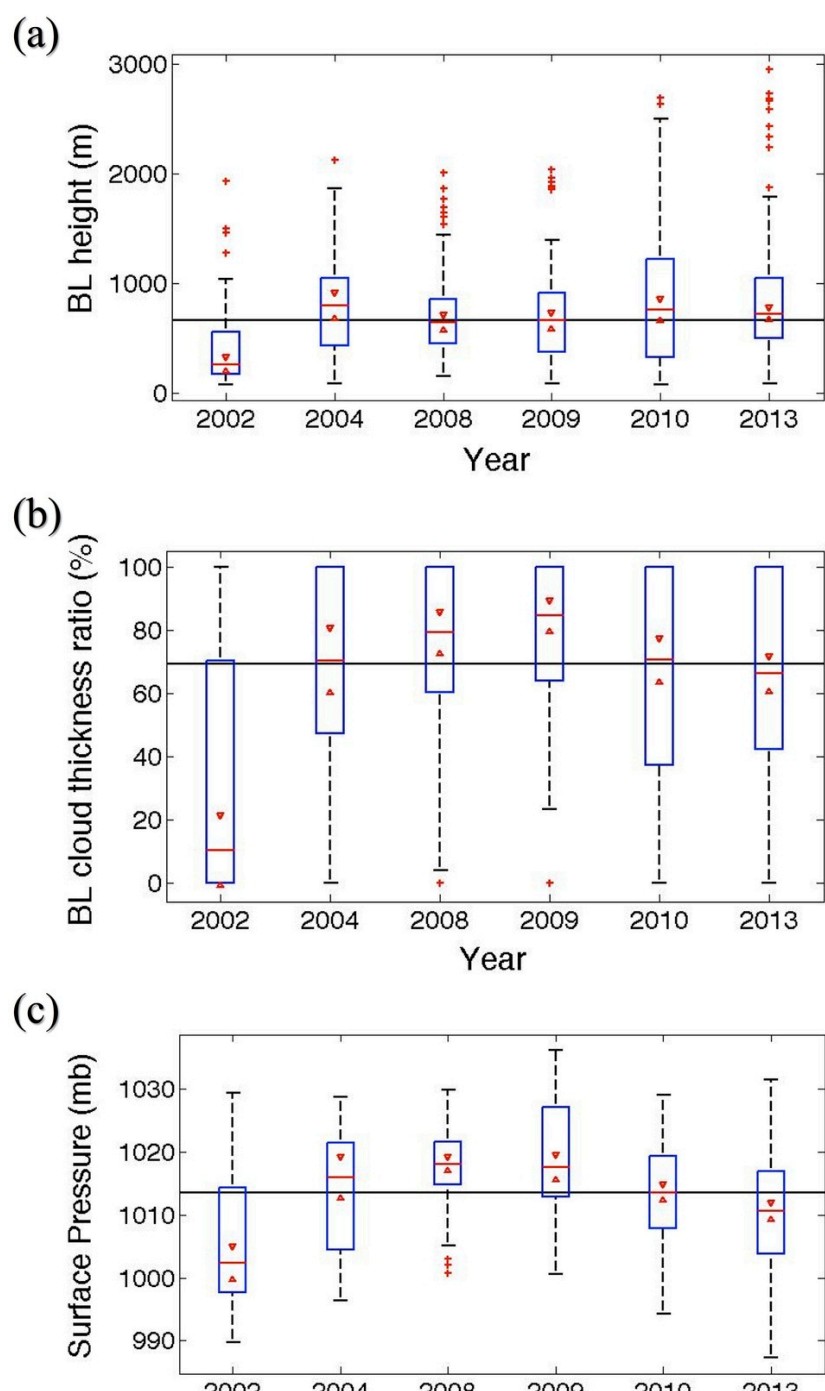

**Figure 8: Same as Fig. 3 but for (a) boundary layer height, (b) boundary layer cloud thickness ratio, and (c) surface pressure.**



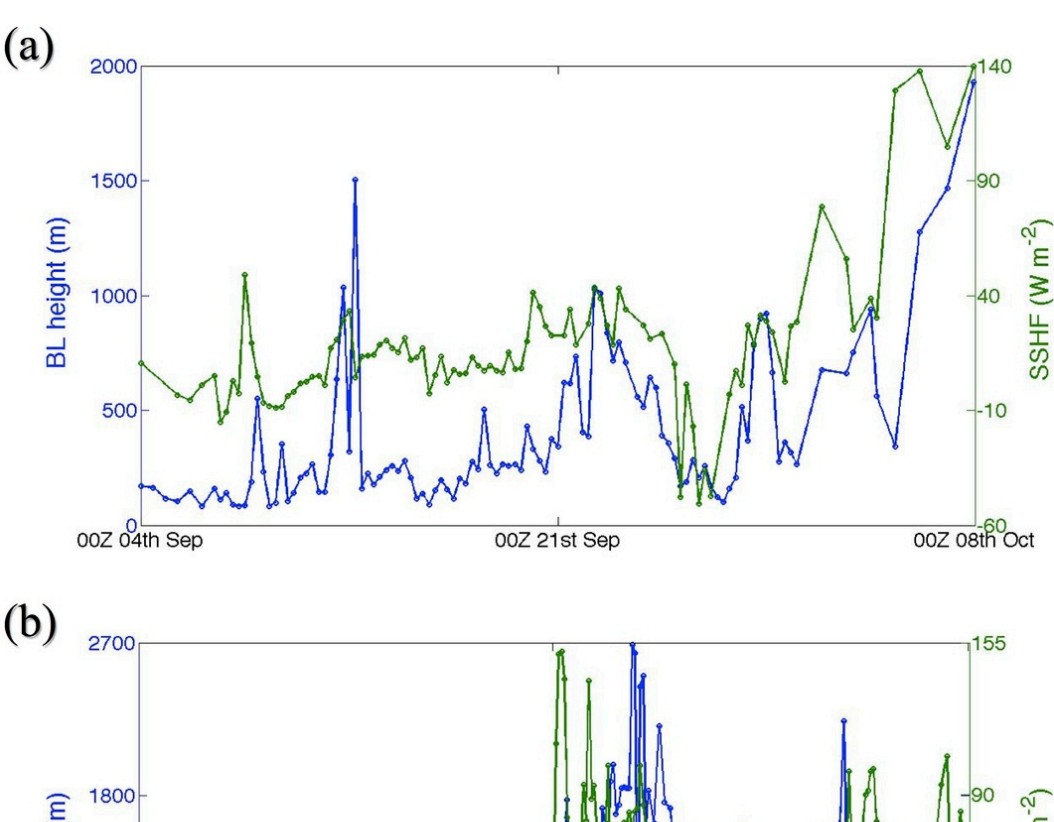

**Figure 9: The time-series of the boundary layer height (blue line; left axis) and the surface sensible heat flux (green line; right axis) during the period of the cruise in (a) 2002 and (b) 2010. The dots represent the time instances of actual measurements.**



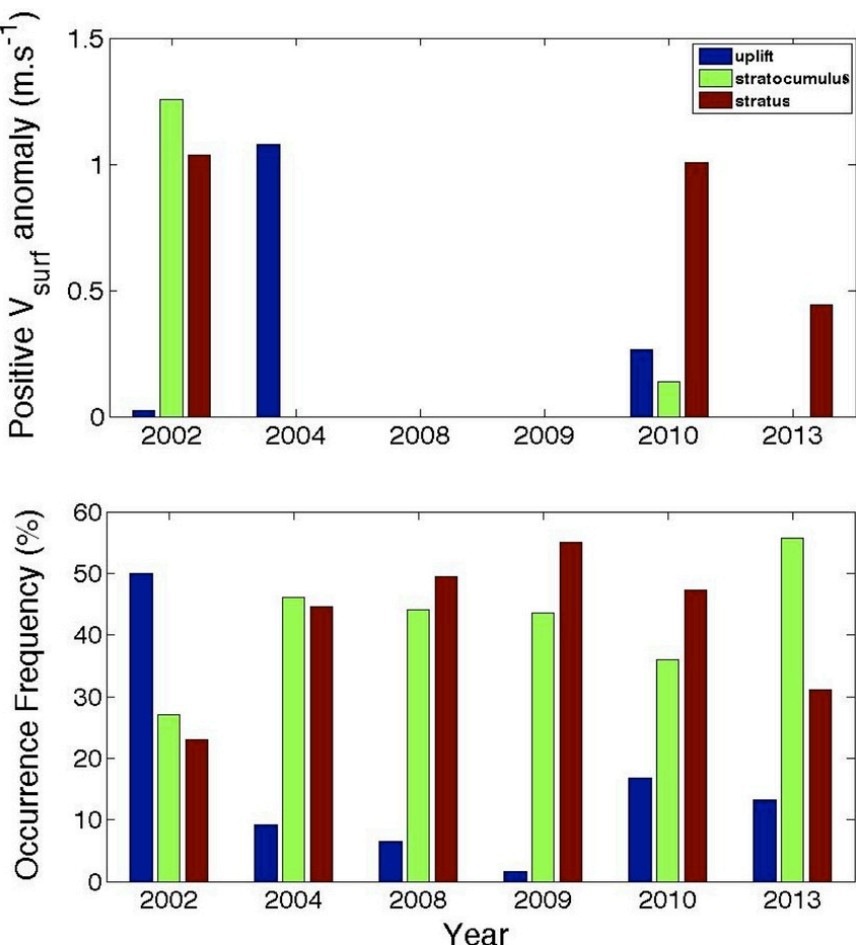

**Figure 10: The distribution of occurrence frequency of the different cloud regimes within a given year (bottom panel), and the wind speed anomaly (positive only) for each year and each regime (top panel). Positive wind speed anomalies are calculated with respect to the mean wind speeds for the uplift, stratocumulus, and stratus regimes, which are 7, 7.2, and 7 ms$^{-1}$, respectively.**

