# Peer review of "The open ocean sensible heat flux and its significance for Arctic boundary layer mixing during early fall"

_Atmospheric Chemistry and Physics, 2015_

## Referee Comment (RC1) · Anonymous Referee #2 · 22 May 2016

This paper explores important relationships between surface sensible heat flux, cloud properties, boundary layer structure, the large scale environment and the surface condition. I commend the authors for attempting to synthesize the unique perspectives coming from the deployment of the R/V Mirai during the fall season. I think that the analysis completed is interesting and sheds some light into how the surface and atmosphere interact. I do have some reservations and questions about the manuscript, and therefore would like to see some additional work completed before this paper is accepted for publication.

Specific Comments: Page 3, line 19: What is the impact of the ship on these measurements?

Section 3.2: I question how well the parcel-based method works when applied in conditions where there may be stratification. Given that the Arctic Ocean surface is open during the analysis times, these are likely times when this technique is generally acceptable. However, I would think that this would not be appropriate in cases where there is a surface inversion, for example, or in instances where clouds have worked to develop temperature structures and are not connected to the surface condition (as is pointed out to occur on occasion in the Arctic).

Page 4, line 30: I struggle with the "boundary layer" terminology as applied. If the layer or cloud associated with it is decoupled, is it still a boundary layer? It seems that the layer may better be referred to as a "decoupled, cloud-driven mixed layer" or similar. I do understand that as defined, the "BL height" may still be located at the cloud height, but perhaps that is also justification for revisiting that definition.

Page 4, line 32: Again, are decoupled clouds really boundary layer clouds? Why would clouds at 1 km be any different than clouds at 3 km if both are decoupled?

Page 5, line 2: In my experience, the near-surface humidity can often be 90% or more. Has an evaluation been completed of the impact of this definition on true cloud statistics? Doesn't the MIRAI also feature surface-based remote sensing? The frequent occurrence of BL cloud thickness ratios of 95% or greater in figure 5 is somewhat concerning. Perhaps it would be appropriate to evaluate the sensitivity of these metrics to the RH threshold chosen (for example, how does fig. 5 change if you choose 97% RH as the threshold?).

Page 5, line 13: I think that I understand this to mean that there were colder SATs observed in the recent years, is that correct? Otherwise can you explain how the variability of the SAT would result in increased surface sensible heat flux? Also, it might be informative to show the components that go into calculating deltaT. For example, how do the SSTs compare between years?

Page 7, line 7-8: Interestingly, this is backwards from what I usually think about Arctic

clouds (thick = frontal, thinner = stratocumulus, thinnest = decoupled stratus). I think it is important to remind the reader that this is cloud thickness within the boundary layer, and not total cloud thickness.

Page 7, line 13: This "(r)" should be positioned after "correlation coefficient", not after "BL height".

Page 7, line 17-18: Yet as a whole, this regime does have deeper boundary layers than the two regimes with smaller deltaT.

Page 7, line 18-19: "indicating that stratocumulus clouds 20 likely form by saturating to the significantly colder air mass that is advected above the surface" — I'm not sure I follow what this means exactly. Suggest rewording for clarity.

Page 7, line 25: More significant in what way?

Page 7, line 31: More significant in what way?

Page 8, line 17: Please redefine what "it" is in this sentence. I believe that you're referring to SSHF, but that should be explicitly stated in the text.

Page 9, line 3: Is there a reason for thinking that the Arctic will see higher wind speeds in a future climate (or a reference which makes a case for this)?

Section 5: I find this section to be less of a discussion, and more of a repetition of already stated findings.

Page 9, lines 24-25: I'm confused — I thought that the higher wind speeds were shown to be a significant factor in the stratus regime, and not in the CAA/stratocumulus regime?

Page 10, lines 10-12: To what extent is this dependent upon the timing of the cruises? Does this number change under as the ocean advances towards refreezing in late October and early November, when air temperatures are colder? It might be nice to include information on the variability in observed SAT between the different years.

[Figure]

Page 11, lines 7-9: I realize that this is supposed to be summarizing the previous text, but this has been stated many times already throughout the manuscript. I would have liked to see some more concrete discussion which synthesizes these results with other studies (without repeating the results of the current study over and over again).

Section 6, bullet points: Again, I feel as though all of this has been stated many times already. I really don't see a need to repeat it a 3rd or 4th time.

Page 12, lines 12-13: What model physics need to be improved? The flux parameterizations? The cloud microphysics and radiation? Ocean dynamics and sea ice physics? More information on these questions would be more helpful than additional repetition of the results of the current study.

Figure 1: I'm not sure that it's necessary to show a map of the entire Arctic here. I think it would help to zoom in on the area of interest (say 60-90 N and 110W to 160 E).

Figure 7: This caption is somewhat confusing. If I understand correctly: - The left hand figure is for uplift regime, and for all wind speeds, and the relationship is derived using all cases except the one outlier. How is this determined to be an outlier? Why are the cases with high BL height and very little SSHF not also outliers? - The right hand figure is for stratus cases, and is divided into two subsets — one for higher wind speeds (red) and one for lower (black). Why is there no relationship determined for the lower wind speeds? Please reword the caption for clarity.

---

## Referee Comment (RC2) · Anonymous Referee #1 · 26 May 2016

A quick report of `The open ocean sensible heat flux and its significance for Arctic boundary layer mixing during early fall' by Ganeshan and Wu.

The paper was well re-written based on my previous comments. However, following issues should be reconsider before acceptance of this manuscript. 1) Analysis using 2013 data, 2) method of cloud fraction estimation, 3) definition of cloud regimes, and 4) interpretation of the relationship between SSHF and SLP. Detail comments and suggestions were shown below.

The data in 2013 cruise:

  According to the reply comment, the authors seemed to try to use 2013 data. The observations were mainly made not only along a single longitude (168W) but also at the fixed point during two weeks (72.75N 168.25W). This period should be very suitable to look at the change in BL height. The observing network during the cruise was described in Inoue et al. (2015), and the information of strong winds and surface heat fluxes was shown in Fig.3 in Kawaguchi et al. (2015). Maximum wind speed was more than 13 m/s, and SSHF during the period was more than 100W/m^2, providing a good sampling range for Figs. 3,4,5,11 and 12 in this manuscript. Please try to use the 2013 data in the analysis and results.

Estimation of cloud fraction:

  The cloud fraction is a very important parameter in this study to decide the cloud regimes. However, the method of estimating the cloud fraction is not well described.

Lines 125-126: The definition of cloud fraction in this study is `the percentage ratio of cloud layer thickness within the BL'. I think that this definition is not the cloud fraction but cloud layering and its vertical integration. Please cite relevant references to prove this method correct. Although the authors cited Sato et al. (2012) to estimate the cloud layer, Sato et al. (2012) used the 90% relative humidity as a threshold to detect the bottom height of cloud layer (they did not estimate the cloud fraction).

Lines: 127-129: Please show the cases of cloud fraction with 30, 50, and 80 %

instead of the extreme case (i.e. 0%). Ceilometer data used in Sato et al. (2012) would be suitable to calculate the hourly and/or daily effective cloud fractions by averaging 1-min data. Visual observations of cloud cover would be also available in the cruise report at radiosonde section. These data would be used to validate the cloud fraction estimated in this study.

Effects of the surface pressure:
The authors pointed out the relationship between SLP and SSHF/BL height. However, SLP is not a suitable physical parameter to modify the SSHF and BL height. Wind speed and temperature advection induced by SLP and temperature gradients should be essential to understand the linkage of SSHF /BL height.

Dry regime:
Lines 210-212: I can not understand the physical relationship between low SLP and low BL cloud fraction. In case of cyclones, the SLP is low with high cloud fraction. What kind of atmospheric situation is considered in the dry regime ?

Medium cloud regime:
Line 214: How did authors confirm the stratocumulus-topped boundary layer ?
Line 220-224: One of the reasons why the correlation of SSHF and BL height was low is that a cloud layer is not always detected by each profile because cloud cover during CAA is relatively low. Therefore, larger variability of temperature in dry regime (e.g. Fig. 10a) came from inclusion of CAA cases.

Line 243: The procedure to estimate the cloud fraction is doubtful, suggesting that the results heavily depend on cloud fraction method. Therefore, the correlation shown here is not so meaningful.

Lines 244:247: By definition, delta T is a main factor for SSHF shown in (Eq. 1). And I do not agree that boundary layer mixing is strongly controlled by

the surface pressure. There is no physical process between BL mixing and surface pressure. The author should mention about wind speeds instead of SLP.

Lines 268-269: Here, the definition of stratus and stratocumulus regimes should be described. Fig. 12 is a very different style comparing with Fig. 8. The authors should mention how to calculate occurrence frequency of three cloud regimes, in particular a threshold between stratus and stratocumulus, i.e. the value of 75%.

Lines 299-304: The authors should note logically that SSHF is not an only factor determine BL height. Do not conclude the story just using the correlation coefficient shown in Table 2. In addition to this, Fig. 7 is meaningless because cyclones induce both of southerly warm advection and northerly cold advection associated with the passage of a frontal system. The former generates negative SSHF with lower BL height, while the latter generates positive SSHF with developed BL, suggesting that the wide range of SSHF and BL height makes the correlation coefficient high. Again, SSHF depends on temperature difference and wind speeds as shown in (Eq. 1). Even in high SLP years, wind speeds were high because the ship might locate at the edge of the Beaufort high. Therefore, the logic which explains the relationship between SSHF and SLP would be totally incorrect.

Other comments:
Lines 325-326: In general, the cloud-driven turbulence is formed by radiative cooling at the cloud top. I can not find such explanation related to cloud-driven turbulence in sections 4.2 and 4.3.3.

Line 333: In general, the cloud cover is very high in case of cyclones.
Again, what kind of atmospheric state is considered under the low surface pressure under clear sky ?

Line 347-352: I think that the threshold value of cloud fraction between

stratus and stratocumulus is very arbitrariness (75%: Table 2). The discontinuous value to divide the frequency distribution in Fig. 8a must be between 85 % and 95 %. Please recalculate the all parameters and figures including correlation coefficients shown in Tables; otherwise I do not agree with the content in this paragraph.

Line 365: Please explain the physical processes related to sea level pressure.

Line 368: What is a low-pressure condition ? A strong cyclone or frequent passages of weak cyclones ? The author should look at the weather charts in 2002.

Line 380: I can not imagine the cloud-free and low-pressure conditions. Is it a real physical condition to explain the results ?

Lines 386-391: At first, the authors should reconsider the method to estimate the cloud fraction.

Reference list:
Line 39: Nicholls and Leighton (1986) is missing from the list.
Line 397: Grønas and Skeie (1999) is missing from the list.
Kalnay et al. (1996) is not cited in the text.
Line 398: Pagowski and Moore(2001) is missing from the list.

Relevant references:
Inoue et al. (2015), Additional Arctic observations improve weather and sea-ice forecasts for the Northern Sea Route. *Sci. Rep.*, 5, 16868.
Kawaguchi et al. (2014), Fixed-point observations of mixed layer evolution in the seasonally ice-free Chukchi Sea: turbulent mixing due to gale winds and internal gravity waves. *J. Phys. Oceanogr.*, 45, 836-853.

---

## Author Comment (AC2) · 19 Aug 2016

Specific Comments:

Comment 1: The method to estimate the cloud fractions should be describe clearly.

Response: In the current manuscript, we have replaced the term "boundary layer cloud fraction" with "boundary layer cloud thickness ratio" which is defined in Section 3.3, as the percentage ratio of cloud layer within the BL. The cloud-base is defined as the first layer above the surface where the relative humidity (RH) equals 90% or more (as in Sato et al. 2012), and the cloud layer is calculated as the vertical integral of all layers within the boundary layer that exceed 90% RH. As an example, for the profile shown in

Fig. 2 (inset) of the revised manuscript, the cloud-base and BL heights are calculated as 580 m and 1180 m respectively, and the BL cloud thickness ratio is estimated to be ~50%.

Comment 2: The definition of cloud regimes were unclear.

Response: The clouds are categorized based on the BL cloud thickness ratio as explained in Section 4.2.1 (Pg. 6, Lines 16-18) of the revised manuscript. Three distinct BL cloud types emerge, with thickness ratios peaking at 5%, 65%, and 95% (Fig. 5 (a)). Consequently, three categories are identified consisting of low (<20%), medium (20-80%), and high (>80%) BL cloud thickness ratios, respectively. Each category corresponds to one (or two) of the four regimes described by Barton et al. (2012). The low BL cloud thickness category (<20% ratio) corresponds to the "uplift regime", the medium BL cloud thickness category (20-80% ratio) corresponds to the "stable" regime, and the high BL cloud thickness category (>80% ratio) corresponds to the "highly stable and very highly stable" regimes described by Barton et al. (2012). This is expressed in Table 1 of the revised manuscript.

Comment 3: The interpretation of the role of SLP on the surface sensible heat flux is questionable. Response: We would like to stress that, in our study, we do not explicitly state that the SLP influences the surface sensible heat flux (SSHF). The SSHF depends on the surface winds and the air-sea temperature gradients as indicated by Eq. 1 of the revised manuscript. However, the efficiency of turbulent mixing might be favored within an unstable boundary layer which often accompanies cyclones or storms. There is some evidence for increased surface turbulent heat transfer during cyclonic activity in the Arctic Ocean (Nilsson et al. 2001, Brummer et al. 1994). Moreover, in the revised manuscript, we discuss the thermodynamic equation for the Arctic BL (Eq. 2) in order to better understand the factors favouring turbulent mixing. It is possible that warm air advection (stratus regime) and low-pressure conditions (uplift regime) will favour rising motion/adiabatic cooling, which may lead to an unstable lapse-rate in the Arctic BL (term C in Eq. 2 of the revised manuscript). This may subsequently lead

to a good correlation between the SSHF and BL height as turbulence is favoured in an unstable BL. Whereas high-pressure or cold air advection typically causes sinking motion in the BL, which may be responsible for the poor correlation between the SSHF and BL height during the stratocumulus regime. (The reader should be cautioned that this is purely speculative and we have not evaluated the relative contributions from thermal advection and compensatory adiabatic motions in the Arctic BL).

References:

Barton, N. P., S. A. Klein, J. S. Boyle, and Y. Y. Zhang, 2012: Arctic synoptic regimes: Comparing domain-wide Arctic cloud observations with CAM4 and CAM5 during similar dynamics, J. Geophys. Res., 117, D15205, doi:10.1029/2012JD017589.

Brümmer, B., Busack, B., Hoeber, H., & Kruspe, G.,1994: Boundary-layer observations over water and Arctic sea-ice during on-ice air flow. Boundary-Layer Meteorology, 68(1- 2), 75-108. Nilsson, E. D., Rannik, Ü., & Håkansson, M., 2001: Surface energy budget over the central Arctic Ocean during late summer and early freezeup. Journal of Geophysical Research: Atmospheres (1984–2012), 106(D23), 32187-32205.

Sato, K., J. Inoue, Y. M. Kodama, & J. E. Overland, 2012: Impact of Arctic sea‐ice retreat on the recent change in cloud‐base height during autumn.Geophysical Research Letters, 39(10).
* * *

---

## Author Response (AR1)

[revised manuscript text omitted]

Thus, although the SSHF presently controls only 10% of the BL height variability, the relationship can become stronger under more frequent occurrences of the uplift regime and/or high surface wind speeds associated with the stratus regime. As a result, the changing patterns of future Arctic climate, such as more frequent storms and greater wind stress (Hakkinen et al. 2008; Smedsrud et al. 2011), may act to enhance the ocean-atmosphere coupling. In the following section, we will further evaluate the relative roles of SSHF and clouds in atmospheric mixing based on boundary layer thermodynamics.

**4.4 Boundary layer thermodynamics over the open Arctic Ocean**

The role of SSHF for BL mixing is better understood by examining the thermodynamic equation (Eq. 2).

$$\int_0^{BL} \frac{\partial T}{\partial t} + \int_0^{BL} \overline{V}\nabla T + \int_0^{BL} (\Gamma_{adiabat} - \Gamma_{env})w = \int_0^{BL} \frac{Q}{c_p}$$

$$\text{(A)} \qquad \text{(B)} \qquad \text{(C)} \qquad \text{(D)}$$

where A = time-rate of change in local temperature

B = horizontal temperature advection in the BL

C = product of the vertical velocity and the difference between the adiabatic and environment lapse-rates

D = diabatic heating/cooling that can be caused by surface turbulent fluxes, cloud (condensational/evaporative) processes, and radiative mechanisms.

The explanation of various terms in Eq. (2) can be found in Holton (2004). The measure of the atmospheric stability (Term C) is zero, by definition, for a well-mixed boundary layer because the environmental lapse-rate ($\Gamma_{env}$) equals the dry/moist adiabatic lapse-rate ($\Gamma_{adiabat}$; see Section 3.2). We inspect processes (sources of vertical motions) that are necessary to make $\Gamma_{env}$ equal to the $\Gamma_{adiabat}$. If we assume that horizontal thermal advection (term B) is homogenous leading to uniform changes in the local temperature profile (term A), then it may not influence the atmospheric stability ($\Gamma_{env}$). However, $\Gamma_{env}$ can change in response to diabatic heating in the Arctic atmosphere (term D), which includes two main sources, viz., surface (sensible and latent) heat fluxes and clouds.

The largely positive temperature gradient ($\Delta T$) during early fall (Fig. 3 (b)) is favourable for episodes of upward sensible heat flux and rising air parcels which can make $\Gamma_{env}$ more unstable. (Note that although the surface latent heat fluxes

are not measured here, they are typically proportional to the SSHF). Similarly, Arctic clouds which form due to large-scale processes (Herman and Goody 1976; Morrison et al., 2012; Solomon et al., 2014) can generate vertical mixing due to a combination of condensational warming and cloud-top radiative cooling. Now, in the absence of low-clouds, the surface fluxes will be the main factor to control $\Gamma_{env}$ and the BL mixing. This is consistent with our results which basically show a good correlation between the SSHF and BL height for the cases with near-zero BL cloud thickness (uplift regime in Table 1 and Fig. 7 (a)). On the other hand, when low-clouds are observed in the vicinity of well-mixed layers (stratus and stratocumulus regimes in Table 1), a weaker relationship is found to exist between the SSHF and BL height, as condensational warming may enhance the height of the surface mixed layer. The shallow BL observed in 2002 is likely due to the lack of occurrence of low-clouds during that year (Fig. 8 (a),(b)).

Now, let us consider the large-scale environment associated with cloudy Arctic BLs. It appears that CAA leads to stratocumulus cloud formation, whereas stratus clouds occur in a warm, subsiding environment (Figs. 5 and 6; Sato et al. 2012). One might expect greater cloud-generated BL turbulence in the latter as they have a lower cloud base and a larger cloud fraction within the well-mixed layer (Table 1 and Fig. 5). Conversely, the SSHF contribution is speculated to be much higher during the stratocumulus regime due to the larger $\Delta T$ associated with CAA (Table 1). But, we find no significant relationship between the BL height and $\Delta T$, as explained in section 4.2. During CAA, it appears that the SSHF contributes very little (if at all) to BL mixing, which suggests that condensational warming due to stratocumulus cloud formation is likely to be more significant. Contrary to our findings, a recent modelling study by Deser et al. (2010) showed that during cold months, CAA over the Arctic Ocean is nullified by the substantial release of surface heat fluxes with minimal contribution from cloud condensational warming. It is therefore important to re-evaluate the role of diabatic heating sources in Arctic climate models, especially their sensitivity to air-sea temperature gradients, by using observational guidance from campaigns like the R/V Mirai. Our study also indicates that the surface fluxes can control BL mixing under strong winds during the warmer stratus regime (Fig. 7(b)).

**5 Discussions**

**5.1 Relationship between the declining sea ice and clouds**

While it is clear that the early fall open Arctic Ocean has a deeper well-mixed boundary layer with more convective clouds compared to its ice-covered counterpart (Sato et al. 2012; Schweiger et al. 2008; Kay and Gettelman 2009), the role of surface heat fluxes remains debatable. Based on our analyses, the SSHF contributes to only 10% of the BL height variability. Moreover, the deepest and most convective BLs (stratocumulus cloud regime) appear to develop independently from the surface and are likely produced by a combination of in-cloud moist and radiative processes supported by large-scale CAA. It is expected that the changing frontal dynamics along the marginal sea ice zone (Kolstad and Bracegirdle 2008), as well as the reportedly enhanced moisture flux over the ice-free ocean (Boisvert and Stroeve 2015; Boisvert et al. 2015), may influence the formation of the deep, well-mixed stratocumulus boundary layer observed during early fall. If this is the case,

Manisha Ganeshan 8/10/2016 8:14 PM

Ganeshan, Manisha…, 8/15/2016 3:01 PM

Manisha Ganeshan 8/11/2016 12:07 PM

Ganeshan, Manish…, 8/10/2016 12:59 PM

Manisha Ganeshan 8/11/2016 12:07 PM

Ganeshan, Manish…, 8/10/2016 12:50 PM

Manisha Ganeshan 8/10/2016 8:15 PM

Ganeshan, Manisha…, 8/15/2016 3:02 PM

Manisha Ganeshan 8/10/2016 8:16 PM

Ganeshan, Manisha…, 8/10/2016 1:01 PM

Ganeshan, Manisha…, 8/10/2016 1:01 PM
**Formatted** ... [28]

Manisha Ganeshan 8/11/2016 5:08 PM

Ganeshan, Manish…, 8/10/2016 12:34 PM

Manisha Ganeshan 8/11/2016 5:08 PM

Ganeshan, Manisha…, 8/10/2016 1:02 PM

Manisha Ganeshan 8/10/2016 8:21 PM

Ganeshan, Manish…, 8/10/2016 12:35 PM

Manisha Ganeshan 8/11/2016 12:13 PM

Ganeshan, Manish…, 8/10/2016 12:36 PM

Manisha Ganeshan 8/14/2016 11:15 PM

Ganeshan, Manisha…, 8/15/2016 3:48 PM

Manisha Ganeshan 8/11/2016 5:14 PM
**Formatted** ... [34]

Manisha Ganeshan 8/11/2016 2:37 PM

Manisha Ganeshan 8/11/2016 5:14 PM
**Formatted** ... [36]

Ganeshan, Manisha…, 8/15/2016 4:45 PM

then the retreating boundaries and the shrinking area of the summer Arctic sea ice, will have a direct influence on the open ocean cloud characteristics which in turn may have significant radiative feedbacks to the darker, ice-free Arctic Ocean. It is therefore imperative to study the changing nature of the large-scale circulation and cloud regimes in relation to the retreating sea ice.

Moreover, some model studies suggest that the surface turbulent fluxes dominate polar amplification during fall and early winter (Bekryaev et al. 2010; Serreze et al. 2009; Deser et al. 2010; Tietsche et al. 2011). Our results suggest that cloud condensational warming and radiative processes might be as important (if not more) for atmospheric warming.

**5.2 Implications for sea ice recovery mechanisms**

During late summer and early fall, the turbulent heat loss from the ocean is considered important for initiating refreeze processes (Inoue and Hori 2011; Tietsche et al. 2011). Our study shows that the efficiency of turbulent heat exchange has not increased substantially over the regions that have recently experienced accelerated summer sea ice loss. Thus, we strongly recommend continuing the exploration of mechanisms that contribute to the cooling of the ocean and the recovery of the fast declining Arctic sea ice.

Model simulations may have a more optimistic representation of the ocean-atmosphere interactions in the dynamic new ice-free Arctic, which appears to be sensitive to the temperature gradient (ΔT) at the surface (Deser et al. 2010; Tietsche et al. 2011; Schweiger et al. 2008). For example, simulations show that the complete loss of summer ice over the Arctic Ocean will be reversed during the following cold season (fall and winter) due to enhanced heat flux from the ocean to the atmosphere during fall (Tietsche et al. 2011). However, more recent measurements have shown that the ocean heat gained during summer can be sustained over the period of fall and winter without being immediately dissipated to the atmosphere, thereby slowing the recovery of sea ice (Jackson et al. 2010; 2012). The chances of possibly irreversible and more permanent feedbacks of sea ice loss need to be seriously evaluated in models. Based on our results, it appears that conditions of uplift and high surface wind speeds may favour efficient heat dissipation by SSHF, whereas episodes of CAA may not. Nilsson et al. (2001) similarly found that the late summer/early fall turbulent heat fluxes over the Atlantic sector of the open Arctic Ocean can be sensitive to cyclone activity and cloud regimes. These dynamical triggers should be duly considered in BL

Ganeshan, Manisha…, 8/15/2016 3:23 PM

Manisha Ganeshan 8/14/2016 10:25 PM

Ganeshan, Manisha…, 8/15/2016 3:23 PM

Manisha Ganeshan 7/28/2016 4:02 PM

Manisha Ganeshan 8/11/2016 3:42 PM

Manisha Ganeshan 8/11/2016 4:16 PM

Ganeshan, Manish…, 8/15/2016 12:38 PM

Ganeshan, Manisha…, 8/15/2016 4:01 PM

Manisha Ganeshan 8/15/2016 7:10 PM

Ganeshan, Manish…, 8/15/2016 12:40 PM

parameterization schemes and surface layer schemes of climate models while evaluating future scenarios and sea ice recovery mechanisms for the Arctic.

**6 Conclusions**

For the rapidly evolving Arctic region, a comprehensive understanding of ocean-atmosphere interactions and underlying coupling processes is crucial for improving regional and global climate models and their predictability. Current models have significant differences in important physical processes such as the efficiency of turbulent heat transfer from the ocean to the atmosphere (Kolstad and Bracegirdle 2008; Grønas and Skeie 1999; Pagowski and Moore 2001). The primary goal of this study was to evaluate and quantify the role of open ocean sensible heat flux in Arctic BL turbulence based on multi-year ship-based observations acquired during early fall. The main conclusions are summarized as follows:

- The surface sensible heat flux during fall is mostly positive owing to the positive ocean-atmosphere temperature differences (ΔT) over the ice-free ocean. Yet, the instantaneous atmospheric response to enhanced fluxes only occurs during specific large-scale (cloud) regimes. It is favored during the uplift (low-pressure) regime (~ 15% of the cases) followed by the stratus cloud (warm subsidence) regime (~ 41% of the cases). Additionally, the ocean heat dissipation is more efficient during episodes of high surface wind speeds compared to increased ΔT.

- Stratus and stratocumulus clouds are frequently observed in the open Arctic boundary layer, prevalent roughly 85% of the time (Fig. 10). The year 2002 is an exception with low BL cloud fraction due to the anomalously low surface pressure conditions. In agreement with previous work (Sato et al. 2012), it is found that stratus clouds are associated with warm air advection and a shallow BL (high stability), whereas stratocumulus clouds result from CAA and have deeper, well-mixed BLs (low stability). Contrary to speculation, the surface generated turbulence is more strongly favored in the former compared to the latter. The instability associated with the stratocumulus-topped BL appears to be caused by cloud-related (moist adiabatic and radiative) processes rather than surface fluxes.

- Consistent with studies over late-summer sea ice (Tjernstrom et al. 2004; Shupe et al. 2013; Morrison et al. 2012), it is evident that the BL height variability in the open Arctic Ocean is also primarily controlled by the dynamical influence of the large-scale circulation and low-level clouds (Fig. 8). The surface sensible heat flux explains only up to 10% of the BL height variability for all cases observed by R/V Mirai. There is pronounced interannual variability in this relationship, which point towards an influence of surface wind speeds and cloud regimes that may act to weaken/strengthen the ocean-atmosphere coupling during early fall.

This study highlights the need for comprehensive in-situ observations to improve model physics for more reliable projections of the coupled Arctic climate and sea ice in the future. Using available surface and upper-air observations from ship cruises, we provide first-hand insights of the optimal conditions for the SSHF contribution to atmospheric mixing. The relevant

**Ganeshan, Manisha…, 8/15/2016 4:02 PM**

**Manisha Ganeshan 8/11/2016 4:45 PM**
Deleted: for the loss of absorbed heat (during summer) to the atmosphere. A recent modeling experiment has shown that the sensible and latent heat fluxes during fall and early winter are crucial for the recovery of sea ice in the future ice-less Arctic (Tietshce et al. 2011). In this study, we investigate the SSHF variability during early fall as well as its role in BL mixing using multi-year ship-based meteorological observations. Based on its contribution to boundary layer turbulence, we are able to estimate the efficiency of turbulent heat exchange between the open ocean and the atmosphere under varying weather (and cloud) regimes. [44]

**Manisha Ganeshan 8/15/2016 2:13 AM**

**Ganeshan, Manisha…, 8/15/2016 4:53 PM**

**Manisha Ganeshan 8/15/2016 2:01 AM**

**Ganeshan, Manisha…, 8/15/2016 4:53 PM**

**Manisha Ganeshan 8/15/2016 2:02 AM**

**Manisha Ganeshan 8/15/2016 2:20 AM**

**Ganeshan, Manish…, 8/15/2016 12:56 PM**

**Manisha Ganeshan 8/15/2016 2:23 AM**

**Ganeshan, Manish…, 8/15/2016 12:56 PM**

**Manisha Ganeshan 8/15/2016 2:16 AM**

**Ganeshan, Manish…, 8/15/2016 12:47 PM**

**Manisha Ganeshan 8/15/2016 2:04 AM**

**Ganeshan, Manish…, 8/15/2016 12:43 PM**

**Manisha Ganeshan 8/15/2016 2:21 AM**

**Ganeshan, Manish…, 8/15/2016 12:57 PM**

**Ganeshan, Manisha…, 8/15/2016 1:30 PM**

coupling mechanisms identified in this study, especially the influence of large-scale circulation patterns (winds, cloud regimes), can be incorporated in current climate models to investigate the consequences of sea ice loss, and the possibility of irreversible 
[revised manuscript text omitted]

**Specific Comments:**

**Comment 1: The method to estimate the cloud fractions should be describe clearly.**

*Response:* In the current manuscript, we have replaced the term "boundary layer cloud fraction" with "boundary layer cloud thickness ratio" which is defined in Section 3.3, as the percentage ratio of cloud layer within the BL. The cloud-base is

10 defined as the first layer above the surface where the relative humidity (RH) equals 90% or more (as in Sato et al. 2012), and the cloud layer is calculated as the vertical integral of all layers within the boundary layer that exceed 90% RH. As an example, for the profile shown in Fig. 2 (inset) of the revised manuscript, the cloud-base and BL heights are calculated as 580 m and 1180 m respectively, and the BL cloud thickness ratio is estimated to be ~50%.

15 **Comment 2: The definition of cloud regimes were unclear.**

*Response:* The clouds are categorized based on the BL cloud thickness ratio as explained in Section 4.2.1 (Pg. 6, Lines 16-18) of the revised manuscript. Three distinct BL cloud types emerge, with thickness ratios peaking at 5%, 65%, and 95% (Fig. 5 (a)). Consequently, three categories are identified consisting of low (<20%), medium (20-80%), and high (>80%) BL

20 cloud thickness ratios, respectively. Each category corresponds to one (or two) of the four regimes described by Barton et al. (2012). The low BL cloud thickness category (<20% ratio) corresponds to the "uplift regime", the medium BL cloud thickness category (20-80% ratio) corresponds to the "stable" regime, and the high BL cloud thickness category (>80% ratio) corresponds to the "highly stable and very highly stable" regimes described by Barton et al. (2012). This is expressed in Table 1 of the revised manuscript.

**Comment 3: The interpretation of the role of SLP on the surface sensible heat flux is questionable.**

*Response:* We would like to stress that, in our study, we do not explicitly state that the SLP influences the surface sensible heat flux (SSHF). The SSHF depends on the surface winds and the air-sea temperature gradients as indicated by Eq. 1 of the revised manuscript. However, the efficiency of turbulent mixing might be favored within an unstable boundary layer which

30 often accompanies cyclones or storms. There is some evidence for increased surface turbulent heat transfer during cyclonic activity in the Arctic Ocean (Nilsson et al. 2001, Brummer et al. 1994). Moreover, in the revised manuscript, we discuss the

thermodynamic equation for the Arctic BL (Eq. 2) in order to better understand the factors favouring turbulent mixing.

It is possible that warm air advection (stratus regime) and **low-pressure** conditions (uplift regime) will favour rising motion/adiabatic cooling, which may lead to an unstable lapse-rate in the Arctic BL (term C in Eq. 2 of the revised manuscript). This may subsequently lead to a good correlation between the SSHF and BL height as turbulence is favoured in an unstable BL. Whereas **high-pressure** or cold air advection typically causes sinking motion in the BL, which may be responsible for the poor correlation between the SSHF and BL height during the stratocumulus regime. (The reader should be cautioned that this is purely speculative and we have not evaluated the relative contributions from thermal advection and compensatory adiabatic motions in the Arctic BL).

10   meteorological observations were compared with multiple instruments for quality and accuracy. Thus, we are sure that they are free from the impact of the ship, making them reliable for use in our study.

**Comment 2: Section 3.2: I question how well the parcel-based method works when applied in conditions where there may be stratification. Given that the Arctic Ocean surface is open during the analysis times, these are likely times**

15   **when this technique is generally acceptable. However, I would think that this would not be appropriate in cases where there is a surface inversion, for example, or in instances where clouds have worked to develop temperature structures and are not connected to the surface condition (as is pointed out to occur on occasion in the Arctic).**

*Response:* Our goal is to identify the sources of convective mixing for the well-mixed Arctic BL, which is found to occur more than 90% of the time in the cruise ship observations used in our study. As the ocean is typically ice-free during this

20   period, the Reviewer is correct in estimating that the conditions of stratification are rare (less than 10% frequency). Therefore, the use of the parcel-based method is generally acceptable. We understand that the reviewer is concerned about cases where the mixed layer is formed due to cloud-generated turbulence, and is decoupled from the surface. In the following paragraph, we will explain how and why such cases are excluded from our analysis.

      We use a *bottom-up* parcel method to identify the boundary layer top, therefore we can safely assume that the well-

25   mixed layers are, in fact, coupled to the *surface*. If the well-mixed layer does not include a cloud, only the surface fluxes are assumed to control the boundary layer development, as explained in Section 4.4 (Pg. 9, Line 25) of the revised manuscript. Whereas in cases where a cloud layer (e.g. stratus or stratocumulus) occurs within the boundary layer, the mixing can be caused due to both surface heat fluxes and cloud-generated turbulence. Even if the latter is dominant, the boundary layer will remain coupled to the surface. This is because the adiabatic temperature profile in the boundary layer, which is a pre-

30   requisite for the parcel-based method, ensures free flow and exchange of heat and momentum with the surface. Thus, we are confident to use the bottom-up parcel-based method as it systematically excludes all decoupled cases while identifying the well-mixed boundary layer top.

**Comment 3: Page 4, line 30: I struggle with the "boundary layer" terminology as applied. If the layer or cloud associated with it is decoupled, is it still a boundary layer? It seems that the layer may better be referred to as a "decoupled, cloud-driven mixed layer" or similar. I do understand that as defined, the "BL height" may still be located at the cloud height, but perhaps that is also justification for revisiting that definition.**

*Response:* As explained in the previous response, we use a bottom-up parcel method and therefore only consider boundary layers that are coupled to the surface. The term "decoupled" was intended to convey the stronger influence of clouds on BL mixing compared to the surface. However, this term contradicts the very definition of a boundary layer which is assumed to always be coupled to the surface. Therefore, the sentence has been modified, and the use of the word "decoupled" is avoided here and elsewhere in the text. Thank you for bringing this misnomer to our attention.

**Comment 4: Page 4, line 32: Again, are decoupled clouds really boundary layer clouds? Why would clouds at 1 km be any different than clouds at 3 km if both are decoupled?**

*Response:* See response to Comments 2 and 3.

**Comment 5: Page 5, line 2: In my experience, the near-surface humidity can often be 90% or more. Has an evaluation been completed of the impact of this definition on true cloud statistics? Doesn't the MIRAI also feature surface-based remote sensing? The frequent occurrence of BL cloud thickness ratios of 95% or greater in figure 5 is somewhat concerning. Perhaps it would be appropriate to evaluate the sensitivity of these metrics to the RH threshold chosen (for example, how does fig. 5 change if you choose 97% RH as the threshold?).**

*Response:* Yes, the R/V Mirai does feature Doppler Radar and Ceilometer observations, but we use radiosonde profiles only to estimate boundary layer and cloud properties. As clouds can have large variability even at the microscale, we use a single source (radiosonde) dataset to study its relationship with BL height, mainly to avoid errors/inconsistencies in spatiotemporal sampling due to the use of multiple data sources.

A RH threshold of 90% is chosen in order to account for ice- or mixed phase clouds which often form under sub-saturated (RH < 100%) conditions in the cold Arctic region. The frequent occurrence of 95% BL cloud thickness in Fig. 5 (a) is not concerning as these cases are likely associated with the persistent stratus fog in the shallow Arctic BL (Nilsson and Bigg 1996). Nevertheless, we examined the sensitivity of the cloud characteristics in Fig. 5 against different RH thresholds (89,90,97%), and found that our results and analysis is robust.

As expected, the BL cloud occurrence frequency decreases when the RH threshold is increased beyond 90%, and vice-versa. The three distinct BL cloud thickness peaks shown in Fig. 5 (a) continue to exist at various RH thresholds (80,90,97%). The major difference is in the skewness of the distribution. For RH threshold of 97% (80%), this leads to a leftward (rightward) shift in the median, resulting in more positive (negative) skewness as compared to Fig. 5 (a). For example, when 97% RH threshold is used, the occurrence frequency of the maximum BL cloud thickness drops from 35 to 12%, whereas that of the minimum BL cloud thickness increases from 12 to 35%. While this does not affect the overall statistics of the stratus cloud regime, it appears to negatively impact the identification of stratocumulus clouds which typically form under colder air temperature conditions. Some cases of cold air advection, with RH > 90% but less than 97%, are (perhaps wrongly) classified as "dry" boundary layers with zero cloud layer thickness. The occurrence frequency of stratocumulus clouds reduces from 44% to ~30%, while that of "dry" BL conditions (uplift regime) increases from 15% to ~35%, which does not agree well with past studies cited in Section 4.2.1. On the other hand, the use of 90% RH threshold yields a reasonable distribution of cloudy vs. "dry" or cloud-free BLs, that agrees with past studies (Barton et al. 2012). Due to a better alignment of the occurrence frequency of cloud regimes with previous literature, and to account for ice or mixed-phase clouds that form under cold, sub-saturated conditions, we deem the use of 90% RH as the appropriate cut-off threshold in our study.

**Comment 6: Page 5, line 13: I think that I understand this to mean that there were colder SATs observed in the recent years, is that correct? Otherwise can you explain how the variability of the SAT would result in increased surface sensible heat flux? Also, it might be informative to show the components that go into calculating deltaT. For example, how do the SSTs compare between years?**

*Response:* Yes, this statement has been clarified to state that more negative SATs lead to the broader SSHF distribution observed in recent years (Pg. 5, Line 11). We didn't include the figures showing the distribution of SSTs and SATs, as they are redundant, in our opinion. But these are shown here for your reference (Figs. 1 and 2).

**Comment 7: Page 7, line 7-8: Interestingly, this is backwards from what I usually think about Arctic clouds (thick = frontal, thinner = stratocumulus, thinnest = decoupled stratus). I think it is important to remind the reader that this is cloud thickness within the boundary layer, and not total cloud thickness.**

*Response:* Thank you for pointing this out. We have made sure to remind the reader that this is the boundary layer cloud thickness here (Pg. 7, Lines 3-5), and elsewhere in the text.

**Comment 8: Page 7, line 13: This "(r)" should be positioned after "correlation coefficient", not after "BL height".**

*Response:* The correction has been made (Pg. 7, Line 9). Thank you for your attention to the details.

**Comment 9: Page 7, line 17-18: Yet as a whole, this regime does have deeper boundary layers than the two regimes with smaller deltaT.**

*Response:* Yes, we have noted this in the text of the revised manuscript (Pg. 7, Lines 13-14).

**Comment 10: Page 7, line 18-19: "indicating that stratocumulus clouds likely form by saturating to the significantly colder air mass that is advected above the surface" ̆A ̃T I'm not sure I follow what this means exactly. Suggest rewording for clarity.**

5  *Response:* This sentence has been modified to read more clearly as follows:

"Figure 6 (a) shows that the temperature anomaly in this regime is maximized between 0.3 to 1.5 km altitudes, indicating that cold air advection occurs *above* the surface, where stratocumulus clouds likely form by the release of latent heat of vaporization". This is reflected in Pg. 7, Lines 15-17 in the new manuscript. Moreover, we have added a new section (Section 4.4) in which we use the thermodynamic equation to better explain how stratocumulus clouds may form during
10  CAA over the open Arctic Ocean.

**Comment 11: Page 7, line 25: More significant in what way? Page 7, line 31: More significant in what way?**

*Response:* For the first case, we have replaced the word significant with the word evident (Pg. 7, Line 21). Studies supporting this sentence (Barton et al. 2012; Taylor et al. 2015) are cited in Pg. 7, Lines 22-23.

For the second case, we mean statistically significant at the 99% confidence level as indicated by Table 1. This has been
15  explicitly mentioned in Pg. 7, Line 27 of the revised manuscript.

**Comment 12: Page 8, line 17: Please redefine what "it" is in this sentence. I believe that you're referring to SSHF, but that should be explicitly stated in the text.**

*Response:* "It" refers to the correlation coefficient ($r$) between the SSHF and the BL height. This has been explicitly stated in the revised manuscript (Pg. 8, Line 15).

20  **Comment 13: Page 9, line 3: Is there a reason for thinking that the Arctic will see higher wind speeds in a future climate (or a reference which makes a case for this)?**

*Response:* Some studies have reported that the Arctic will experience more frequent cyclonic conditions and higher wind stress in the future (Hakkinen et al. 2008; Higgins and Cassano 2009; Smedsrud et al. 2011). These have been duly cited in the text (Pg. 8, Line 32).

25  **Comment 14: Section 5: I find this section to be less of a discussion, and more of a repetition of already stated findings.**

*Response:* Section 5 has been re-written as a discussion of our present findings in context with previous literature. Relevant citations such as Boisvert and Stroeve (2015), Boisvert et al. (2015),  Brümmer (1999), Brümmer and Pohlmann (2000), Hartmann et al. (1999), Deser et al. (2010), Higgins and Cassano (2009), Jackson et al. (2010; 2012), Nilsson et al. (2001),
30  are now included in the revised manuscript.

**Comment 15: Page 9, lines 24-25: I'm confused ̆A ̃T I thought that the higher wind speeds were shown to be a significant factor in the stratus regime, and not in the CAA/stratocumulus regime?**

*Response:* Actually, the correlation coefficient '*r*' also improves with wind speeds in the CAA regime though it remains insignificant (Pg. 7, Lines 28-30). Nevertheless, we have removed this sentence as it was too speculative.

**Comment 16: Page 10, lines 10-12: To what extent is this dependent upon the timing of the cruises? Does this number change under as the ocean advances towards refreezing in late October and early November, when air temperatures are colder? It might be nice to include information on the variability in observed SAT between the different years.**

*Response:* We have carried out extensive analyses inspecting the spatial and temporal dependence of the SSHF-BL height relationship, and find that there is no sensitivity to the occurrence of negative SATs. Given that we have observations in October and September, we have looked at monthly differences and there is no evident seasonality. We are confident in our finding that the SSHF control of the BL height (~10%) is independent of the seasonal variations in ΔT. The interannual variability of observed SATs has been included here for your reference (Fig. 2).

**Comment 17: Page 11, lines 7-9: I realize that this is supposed to be summarizing the previous text, but this has been stated many times already throughout the manuscript. I would have liked to see some more concrete discussion which synthesizes these results with other studies (without repeating the results of the current study over and over again).**

*Response:* This entire section has been re-written (section 5 in the new manuscript), to include a more holistic discussion rather than repetition of our findings. New studies are cited (see response to Comment 14) to better synthesize our results with previous literature.

**Comment 18: Section 6, bullet points: Again, I feel as though all of this has been stated many times already. I really don't see a need to repeat it a 3rd or 4th time.**

*Response:* We have modified the section title to "Summary" as opposed to "Conclusions", and the bullet points now include only the key take-away points from our study.

**Comment 19: Page 12, lines 12-13: What model physics need to be improved? The flux parameteri- zations? The cloud microphysics and radiation? Ocean dynamics and sea ice physics? More information on these questions would be more helpful than additional repetition of the results of the current study.**

*Response:* In the revised manuscript, we have included a more comprehensive discussion of the implications of our results for model physics (Pg. 10, Lines 29-32, Pg. 11, Lines 1-2, and Pg. 11, Lines 15-21). This is provided below for your reference.

"Clouds, in turn, can have significant radiative feedbacks to the darker, ice-free ocean surface. Some model simulations of an ice-free Arctic Ocean suggest that the surface heat fluxes will dominate polar amplification during fall and early winter (Deser et al. 2010; Tietsche et al. 2011; Higgins and Cassano 2009). Our results suggest that both surface fluxes and clouds are sensitive to non-local large-scale factors, which influence their relative roles in diabatic heating of the Arctic atmosphere. This interaction between dynamic and thermodynamic variables must be duly incorporated in climate models for accurate projections of polar amplification".

"Based on our results, it appears that conditions of uplift and high surface wind speeds may favour efficient heat dissipation by SSHF, whereas episodes of CAA may not. Nilsson et al. (2001) similarly found that the late summer/early fall turbulent heat fluxes over the Atlantic sector of the open Arctic Ocean can be sensitive to cyclone activity and cloud regimes. These dynamical triggers should be duly considered in BL parameterization schemes and surface layer schemes of climate models while evaluating future scenarios and sea ice recovery mechanisms for the Arctic. The chances of possible irreversible and more permanent feedbacks of sea ice loss also need to be seriously evaluated in models".

**Comment 20: Figure 1: I'm not sure that it's necessary to show a map of the entire Arctic here. I think it would help to zoom in on the area of interest (say 60-90 N and 110W to 160 E).**

*Response:* Figure 1 has been revised as per your recommendation. We have now zoomed in an area covering 120W to 150E for more clarity. Thank you for the suggestion. The new figure is included here for your reference (Fig. 3).

**Comment 21: Figure 7: This caption is somewhat confusing. If I understand correctly: - The left hand figure is for uplift regime, and for all wind speeds, and the relationship is derived using all cases except the one outlier. How is this determined to be an outlier? Why are the cases with high BL height and very little SSHF not also outliers? - The right hand figure is for stratus cases, and is divided into two subsets — one for higher wind speeds (red) and one for lower (black). Why is there no relationship determined for the lower wind speeds? Please reword the caption for clarity.**

*Response:* As explained in section 4.2 (Pg. 5, Lines 25-28), the $\Delta T$ has no influence on the correlation between SSHF and BL height ($r$), whereas surface wind speeds have a positive influence on '$r$'. For the observations of uplift regime shown in Fig. 7 (a), the point identified as an "outlier" has the maximum $\Delta T$ (6.77°C), which does not improve the linear relationship between SSHF and BL height. Whereas the red markers in Fig. 7 (b) are observations with maximum wind speeds, which has a positive influence on the linear relationship. For clarity, the caption has been modified to read as follows:

[revised manuscript text omitted]

---

## Author Response (AR2)

**Overall:**

**The authors have revised the manuscript very well based on my previous comments. The content would be suitable**
5 **for the publication of ACP with very minor revision.**

**Minor comments:**

**I have an opinion of a role of SSHF on BL development. Although the authors focused on the contribution of SSHF (air-sea temperature difference and wind speeds) on the BL developments, SSHF also contributes to heating in the**
10 **BL. We see the phrase `SSHF only explains up to 10 % of the BL height variability' at several places (e.g. P1L19, P5L24, P8L29, P10L22, and P12L13), however, I think that this finding is exaggerated because there is no explanation of BL heating. I encourage them to mention this point explicitly in the manuscript.**

**Response:** We thank the Referee for his comments, and have duly considered his suggestion. The SSHF contribution to BL
15 development is *via* diabatic heating, as explained in Pg. 9, Lines 13-19. The diabatic heating may lead to direct increases in surface air temperatures if the BL is stable, however, as the BL is well-mixed, we have to consider the changes to the temperature profile due to diabatic heating. Therefore, we do not believe that the BL development due to SSHF and the BL heating due to SSHF are independent processes. In any case, we have removed the phrase "SSHF only explains up to 10 % of the BL height variability" from the discussion and summary sections in Pg. 10 and Pg. 12. We have additionally clarified
20 this point in Sections 4.4, 5.1, and 6.

[revised manuscript text omitted]